# VQ-Seg: Vector-Quantized Token Perturbation for Semi-Supervised Medical Image Segmentation

**Sicheng Yang**[1*]  **Zhaohu Xing**[1*]  **Lei Zhu**[1,2†]

[1]The Hong Kong University of Science and Technology (Guangzhou)
[2]The Hong Kong University of Science and Technology

## Abstract

Consistency learning with feature perturbation is a widely used strategy in semi-supervised medical image segmentation. However, many existing perturbation methods rely on dropout, and thus require a careful manual tuning of the dropout rate, which is a sensitive hyperparameter and often difficult to optimize and may lead to suboptimal regularization. To overcome this limitation, we propose VQ-Seg, the first approach to employ vector quantization (VQ) to discretize the feature space and introduce a novel and controllable Quantized Perturbation Module (QPM) that replaces dropout. Our QPM perturbs discrete representations by shuffling the spatial locations of codebook indices, enabling effective and controllable regularization. To mitigate potential information loss caused by quantization, we design a dual-branch architecture where the post-quantization feature space is shared by both image reconstruction and segmentation tasks. Moreover, we introduce a Post-VQ Feature Adapter (PFA) to incorporate guidance from a foundation model (FM), supplementing the high-level semantic information lost during quantization. Furthermore, we collect a large-scale Lung Cancer (LC) dataset comprising 828 CT scans annotated for central-type lung carcinoma. Extensive experiments on the LC dataset and other public benchmarks demonstrate the effectiveness of our method, which outperforms state-of-the-art approaches. Codes will be released[1].

## 1 Introduction

Medical image segmentation serves as a fundamental step in numerous clinical applications, such as disease diagnosis [1–3], anatomical structure delineation [4–6], lesion localization [7–9], and surgical planning [10–12]. In recent years, supervised deep learning methods have demonstrated outstanding performance in segmentation tasks, significantly surpassing traditional approaches in both accuracy and robustness [13–15]. However, these methods typically require large amounts of finely annotated medical data, the collection of which is not only expensive and labor-intensive but also demands substantial domain expertise. To address this limitation, semi-supervised learning, which leverages a small set of labeled data alongside a larger pool of unlabeled samples, has emerged as a promising direction and is receiving growing attention in the medical imaging community.

Consistency learning represents a widely adopted paradigm in semi-supervised medical image segmentation, designed to enforce prediction invariance under various perturbations [16]. Feature-level dropout [17, 16] is a commonly employed technique within this framework, introducing perturbation into intermediate representations to enhance model robustness. However, its effectiveness is critically dependent on the selection of the dropout rate (DR), a hyperparameter.

---

[*]Equal contribution.
[†]Lei Zhu (leizhu@ust.hk) is the corresponding author.
[1] `https://github.com/script-Yang/VQ-Seg`

39th Conference on Neural Information Processing Systems (NeurIPS 2025).

As depicted in Fig. 1, our experiments reveal two primary challenges associated with this methodology. Firstly, the adoption of lower dropout rates (*e.g.*, DR=0.3, DR=0.5) yields negligible impact on segmentation performance. This suggests that the induced perturbation is insufficient to provide meaningful regularization. Secondly, elevating the dropout rate (*e.g.*, DR ≥ 0.7) leads to a rapid decline in performance metrics. Specifically, Dice and Jaccard scores exhibit a sharp decrease, while HD95 and ASD values significantly increase, indicating a substantial degradation in both structural accuracy and boundary delineation. Qualitative analyses further corroborate these findings, demonstrating that segmentation outputs under high dropout rates frequently fail to yield meaningful segmentation results, rendering them practically unusable.

These observations underscore the inherent difficulty in identifying an optimal dropout rate that consistently enhances performance while mitigating the risk of model collapse. Consequently, there is a pressing need for a more stable perturbation strategy, thereby achieving the desired regularization effect in a predictable and robust manner.

In this paper, we introduce VQ-Seg, a novel semi-supervised framework for medical image segmentation. The core innovations of our approach include: first, the Quantized Perturbation Module (QPM) designed to replace traditional dropout by enabling controlled and structured perturbations of encoded features within a discrete VQ space. Unlike dropout, which relies on hyperparameters such as the dropout rate, QPM leverages distances between codebook codewords to define perturbation strategies, thereby offering enhanced interpretability and stability. To address potential visual information loss arising from vector quantization, we tackle this issue from two perspectives. Initially, we construct a dual-branch architecture that shares a post-quantized space, unifying image reconstruction and semantic segmentation tasks within a joint optimization framework in a discrete representation space. This design facilitates the preservation of critical structural information from images while also utilizing the reconstruction task as a self-supervisory signal to encourage the VQ encoder to learn improved representations. Furthermore, to enhance semantic consistency and mitigate the loss of high-level semantic information during quantization, we incorporate a foundation model-guided alignment strategy. Specifically, we develop a Post-VQ Feature Adapter (PFA) that employs contrastive learning to align quantized features with semantic features derived from a pre-trained visual foundation model. This approach

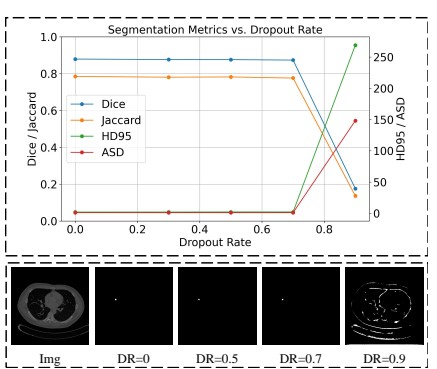

Figure 1: Effect of dropout rate on segmentation performance in a fully supervised setting on the LC dataset. Low dropout rates show negligible impact, whereas a high dropout rate (DR ≥ 0.7) severely degrades both quantitative metrics and visual outputs. Notably, DR = 0.9 leads to unusable predictions, highlighting the challenge of selecting an optimal dropout rate.

effectively enriches the semantic content and spatial consistency of discrete representations. Extensive experiments conducted on our collected Lung Cancer (LC) dataset (comprising 828 annotated cases) and the open-source ACDC dataset demonstrate that VQ-Seg significantly outperforms state-of-the-art semi-supervised segmentation methods across key evaluation metrics, including Dice, Jaccard, HD95, and ASD. Detailed ablation studies further validate the efficacy and synergistic effects of the individual components of VQ-Seg.

In summary, our contributions are five-fold:

- We collected a new large-scale dataset, the Lung Cancer (LC) dataset, comprising 828 chest CT scans with annotations of central-type carcinoma of the lung.

- We propose a novel Quantized Perturbation Module (QPM) to perturb features within a discrete vector quantization (VQ) space. QPM enables a more structured and controllable mechanism for representation perturbation by shuffling the spatial locations of codebook indices, offering enhanced interpretability and stability compared to traditional dropout.

- We introduce a dual-branch architecture where the post-quantized space is concurrently utilized for both image reconstruction and the downstream segmentation task. This design

uses reconstruction as a self-supervisory signal, encouraging the VQ encoder to learn better representations and preserving essential visual information.

- We develop a foundation model-guided alignment strategy, where a frozen foundation model (FM) serves as an external semantic prior to guide and regularize the Post-VQ representation. A Post-VQ Feature Adapter (PFA) is introduced to transform the quantized codebook embeddings into a semantically aligned space using contrastive learning, mitigating the loss of high-level semantic features.

- We compare our method with multiple cutting-edge methods on the LC dataset and open-source datasets, demonstrating that our approach achieves state-of-the-art performance, which verifies its effectiveness.

## 2 Related work

### 2.1 Semi-supervised Medical Image Segmentation

Obtaining large-scale, high-quality manual annotations for medical images is both time-consuming and labor-intensive [18, 19]. Semi-supervised learning (SSL) methods have thus gained attention as a promising solution for medical image segmentation under limited annotation settings [20]. Among various SSL strategies, pseudo-labeling-based approaches are widely adopted due to their simplicity and ease of implementation [21]. These methods [22–25] typically involve training an initial model on the labeled dataset and then generating pseudo-labels. However, the use of pseudo-labels can introduce noise and lead to training instability, especially when incorrect labels are propagated [26]. To address this, subsequent research [27–29] has incorporated consistency regularization to enforce prior constraints on the learned representations. Consistency regularization is grounded in the smoothness assumption, which posits that small perturbations to the input data should not significantly alter the model's predictions [30]. In this context, the design and implementation of perturbations play a critical role in determining the effectiveness of the model.

### 2.2 Feature-level Dropout for Consistency Regularization

We focus on feature-level Dropout strategies. Leveraging Dropout-based regularization for consistency learning has become a widespread approach in semi-supervised medical image segmentation [31]. Specifically, techniques such as Monte Carlo Dropout (MC-Dropout) [16] have been employed to model uncertainty and enforce prediction consistency under random feature perturbations. These perturbations help improve the generalization ability of the model by encouraging it to make stable and robust predictions despite the uncertainty inherent in the data. Several studies [16, 32, 29] have shown the effectiveness of such methods, demonstrating improved performance in semi-supervised medical image Segmentation tasks. However, while these approaches have shown promise, they typically require the careful tuning of a hyperparameter—the Dropout rate. The appropriate setting of the Dropout rate is critical. Both peer observations [29] and our own empirical studies (see Fig. 1) suggest that this process is often challenging in practice, as the optimal Dropout rate may vary depending on the specific dataset, task, and network architecture. In particular, inappropriate Dropout configurations can lead to unstable training dynamics, such as excessive regularization that prevents the model from learning effectively [33, 31]. This highlights a significant limitation of traditional Dropout-based methods. This situation motivates the need for alternative strategies that can provide controlled and effective feature-level perturbations.

## 3 Methodology

### 3.1 Overview

Fig. 2 provides an overview of our Vector Quantization-based Semi-supervised Segmentation framework (VQ-Seg). The input image is first encoded into continuous latent features, which are then quantized into a discrete codebook space via the Vector Quantization (VQ) module. To introduce structured perturbations for consistency learning, we propose the Quantized Perturbation Module (QPM), which shuffles codebook indices based on their learned distances, offering a stable and interpretable alternative to dropout. To compensate for potential visual information loss, VQ-Seg

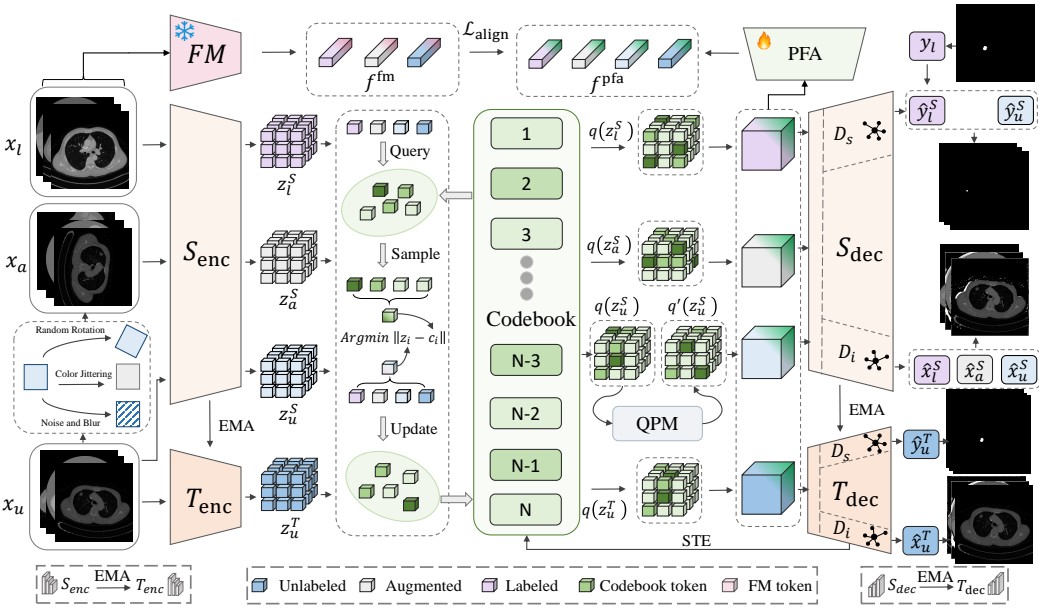

Figure 2: Overview of the VQ-Seg framework. The input image $x$ is encoded into continuous features $z$, which are then quantized into a discrete codebook space via vector quantization (VQ). Quantized Perturbation Module (QPM) introduces controllable perturbations for consistency learning. The dual-branch architecture jointly optimizes image reconstruction and segmentation using the shared Post-VQ features. Additionally, a Post-VQ Feature Adapter (PFA) aligns the quantized features with semantic embeddings from a foundation model (FM).

adopts a dual-branch architecture that jointly optimizes reconstruction and segmentation tasks using the shared Post-VQ feature space. Furthermore, the Post-VQ Feature Adapter (PFA) aligns quantized features with semantic embeddings from a pre-trained foundation model through patch-wise contrastive learning, enriching representation semantics and reducing drift.

## 3.2 Theoretical Motivation

To analyze the sensitivity of the model to perturbations in the feature space, we interpret the KL divergence $\mathrm{KL}(P\|Q)$ as a measure of the perturbation radius from the original distribution $P$ to the perturbed one $Q$. This is inspired by distributionally robust optimization (DRO) [34, 35], where the worst-case risk is evaluated over an uncertainty set defined by a bounded KL divergence. Thus, the divergence reflects the extent to which the input perturbation has structurally shifted the feature representation. Supported by recent studies [36], for Dropout, the KL divergence between the posterior distribution $Q$ (induced by dropout) and a prior distribution $P$ can be approximated as:

$$D_{KL}(P\|Q) \approx \frac{1}{2}\left(\frac{p}{1-p} + \log(1-p)\right),\tag{1}$$

where $p \in (0,1)$ denotes the dropout rate. As $p$ increases, the approximation indicates a growing perturbation radius, with the KL divergence rising sharply (see Appendix A for a full derivation). Such behavior reveals the inherent instability of dropout from a theoretical standpoint: a large dropout rate causes the posterior distribution to deviate significantly from the prior, potentially leading to over-regularization and degraded learning performance, as supported by our empirical results (see Fig. 1). To mitigate this issue, we propose the Quantized Perturbation Module (QPM), which perturbs features within a discrete vector quantization (VQ) space, enabling a more structured and controllable mechanism for representation perturbation.

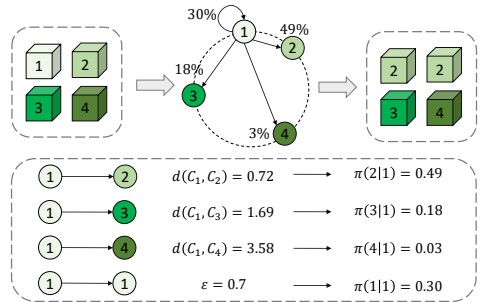
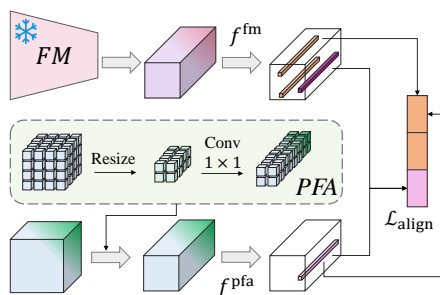

Figure 3: A concrete example of the Quantized Perturbation Mechanism (QPM) with a codebook size of $K = 4$ and a perturbation strength $\epsilon = 0.7$. It illustrates the probabilistic transitions from the original codeword $c_1$ (index 1) to itself and other codewords ($c_2, c_3, c_4$) with their respective probabilities $\pi(j|1)$, where the transition to $c_2$ (49%) exhibits the highest probability of replacement.

Figure 4: Architecture of the Post-Quantization Feature Adapter (PFA) designed for aligning post-quantization features with a frozen Foundation Model (FM) via a patch-wise contrastive loss, $\mathcal{L}_{\text{align}}$. The PFA initially employs a resizing operation followed by a $1 \times 1$ convolution to match the spatial resolution and channel dimensionality of the FM features, thereby facilitating subsequent semantic alignment.

## 3.3 Quantized Perturbation Module (QPM)

In our method, the encoder output $z = f_{\text{enc}}(x)$ represents a continuous feature embedding of the input medical image $x$. This embedding is then projected into a discrete latent space by selecting the nearest codeword from a learnable codebook $\mathcal{C} = \{c_1, c_2, ..., c_K\}$, where $K$ denotes the total number of codewords, *i.e.*, the size of the codebook. The quantized codeword index $i$ is obtained by

$$i = \arg \min_{j \in \{1,...,K\}} \|z - c_j\|. \tag{2}$$

After the encoding step, we apply a perturbation strategy $\pi(j \mid i)$. For each original codeword $c_i$ (with index $i$), a new codeword $c_j$ (with index $j$) is sampled based on the conditional probability $\pi(j \mid i)$, and $c_j$ replaces the original codeword as the input to the decoder. This procedure introduces a structured perturbation within the discrete latent space, resulting in a more controlled and interpretable form of regularization. To implement this perturbation, we first define a prior distribution $P(c_i)$ over the codebook $\mathcal{Z}$. We assume a uniform distribution over all codewords:

$$P(c_i) = \frac{1}{K}, \quad \forall i \in \{1, 2, ..., K\}. \tag{3}$$

The perturbation strategy defines the conditional probability of transitioning from the current codeword $c_i$ to another codeword $c_j$:

$$\pi(j \mid i) = \begin{cases} 1 - \epsilon, & \text{if } j = i \\ \frac{\epsilon \exp(-d(c_i, c_j))}{Z_i}, & \text{if } j \neq i \end{cases} \tag{4}$$

where $\epsilon \in [0, 1]$ is a control term controlling the perturbation strength, $d(c_i, c_j)$ is a distance metric between codewords $c_i$ and $c_j$, and $Z_i = \sum_{k \neq i} \exp(-d(c_i, c_k))$ is a normalization factor. The resulting perturbed distribution $Q(c_j|\epsilon)$ over the codewords is then given by:

$$Q(c_j|\epsilon) = \sum_{i=1}^{K} P(c_i)\pi(j \mid i) = \frac{1 - \epsilon}{K}\delta_{ij} + \frac{\epsilon}{K} \sum_{i \neq j} \frac{\exp(-d(c_i, c_j))}{\sum_{k \neq i} \exp(-d(c_i, c_k))}. \tag{5}$$

The KL divergence between the prior distribution $P$ and the perturbed distribution $Q$ is:

$$D_{KL}(P||Q) = \sum_{j=1}^{K} P(c_j) \log \left( \frac{P(c_j)}{Q(c_j|\epsilon)} \right) = -\frac{1}{K} \sum_{j=1}^{K} \log \left( K Q(c_j|\epsilon) \right). \tag{6}$$

Compared to Dropout, our QPM offers several advantages. The perturbed distribution $Q(c_j|\epsilon)$ is always well-defined and bounded, ensuring numerical stability (Detailed proof can be found

in Appendix B). QPM is directly controlled via a single control term $\epsilon$ and is influenced by the learned codebook structure. Moreover, QPM introduces structured perturbations by probabilistically transitioning between learned codebook entries based on their distances, yielding a potentially more interpretable and controllable form of regularization compared to the stochastic perturbations inherent in Dropout (refer to an example presented in Fig. 3). The perturbation remains within the learned discrete latent space, offering a distinct approach to regularization.

### 3.4 Dual-Branch Architecture with Shared Post-VQ Space

Visual information inherently exists in a continuous space. The process of quantization, while enabling discrete representations, can lead to a loss of fine-grained details and potentially reduce the representational capacity for modeling intricate visual structures [37, 38]. To address this, we introduce a dual-branch architecture where the post-quantization feature space (Post-VQ Space), is concurrently utilized for both image reconstruction and the downstream segmentation task. By jointly optimizing the quantized feature space with respect to these two objectives, we encourage it to encode fundamental structural information, as well as segmentation-relevant representations. This dual utilization facilitates the preservation of essential visual information within the Post-VQ Space, without compromising the performance of the segmentation task.

As depicted in Fig. 2, both the teacher ($T$) and student ($S$) decoder networks adopt this design. Following the encoding stages and vector quantization, we obtain the discrete representations $q(z_l^S)$ for $x_l$, $q(z_a^S)$ for $x_a$, $q(z_u^S)$ for $x_u$ processed by the student, and $q(z_u^T)$ for $x_u$ processed by the teacher. The QPM-perturbed quantized representation of the unlabeled data is denoted as $q'(z_u^S) = $ QPM $(q(z_u^S))$, which is used by the student network. These discrete representations, where $q(\mathbf{z}) \in \{q(z_l^S), q(z_a^S), q(z_u^S), q'(z_u^S)\}$, serve as the input to the image decoder ($D_i$) and the segmentation decoder ($D_s$), as described by:

$$\hat{x} = D_i(q(\mathbf{z})), \hat{y} = D_s(q(\mathbf{z})). \tag{7}$$

The loss function for the labeled data $x_l$ with its associated ground truth segmentation $y_l$ is defined as:

$$\mathcal{L}_l = \mathcal{L}_{rec}(x_l, \hat{x}_l^S) + \mathcal{L}_{seg}(y_l, \hat{y}_l^S). \tag{8}$$

For the unlabeled data $x_u$, we employ a pseudo-labeling strategy. Initially, a predicted segmentation map is generated by the teacher network $T_{\text{dec}}$ as $\mathbf{y}_u^T = D_s^T(q(z_u^T))$, with a corresponding reconstructed image $\hat{x}_u^T = D_i^T(q(z_u^T))$. Subsequently, a pseudo-label $\tilde{y}_u$ is derived from $\mathbf{y}_u^T$ by selecting the class with the maximum probability (argmax operation). This pseudo-label $\tilde{y}_u$ is then used as the target segmentation for the QPM-perturbed quantized representation $q'(z_u)$ of the unlabeled data in the student network $S_{\text{dec}}$. The loss for the unlabeled data in the student network $S_{\text{dec}}$ integrates both the reconstruction loss of the original unlabeled data and a segmentation loss based on the teacher-generated pseudo-label applied to the perturbed representation:

$$\mathcal{L}_u = \mathcal{L}_{rec}(x_u, \hat{x}_u^S) + \mathcal{L}_{seg}(\tilde{y}_u, \hat{y}_u^S) + \mathcal{L}_{seg}(\tilde{y}_u, \hat{y}_a^S). \tag{9}$$

The overall dual-branch loss is defined as:

$$\mathcal{L}_{\text{db}} = \mathcal{L}_l + \lambda_u \mathcal{L}_u \tag{10}$$

where $\lambda_u$ is a hyperparameter that balances the contribution of the unlabeled loss $\mathcal{L}_u$ relative to the labeled loss $\mathcal{L}_l$. $\mathcal{L}_{rec}$ denotes the $L_1$ loss, and $\mathcal{L}_{seg}$ represents the Cross-Entropy loss. By jointly minimizing $\mathcal{L}_{db}$, we encourage the shared quantized feature space to encode information beneficial for both reconstructing the input image and segmenting relevant structures. This synergistic optimization leverages supervision from labeled data and self-supervisory signals from unlabeled data via pseudo-labeling, where the teacher network generates pseudo-labels to guide the student network's learning on the QPM-perturbed quantized representations of the unlabeled data.

### 3.5 Foundation Model-Guided Alignment for Post-VQ Space

Although vector quantization (VQ) compacts the feature space, its discretization process introduce semantic bias and loss of fine details [39], which is particularly detrimental in high-precision tasks such as medical image segmentation. To mitigate these issues, we propose a foundation model-guided

alignment strategy, where a frozen foundation model (FM) serves as an external semantic prior to guide and regularize the Post-VQ representation. Specifically, we introduce a Post-VQ Feature Adapter (PFA) that transforms the quantized codebook embeddings into a semantically aligned space, as illustrated in Fig. 4. The PFA first resizes the VQ features and then applies a $1 \times 1$ convolution to match the spatial resolution and channel dimension of the FM features. Denoting the output of the PFA as $f^{\text{pfa}} \in \mathbb{R}^{H' \times W' \times C'}$ and the corresponding FM features as $f^{\text{fm}}$. We then apply a patch-wise contrastive learning objective [40, 41] to minimize the semantic discrepancy between the adapted VQ features and the FM representations:

$$\mathcal{L}_{\text{align}} = -\frac{1}{HW} \sum_{i=1}^{HW} \log \frac{\exp\left(\text{sim}(f_i^{\text{pfa}}, f_i^{\text{fm}})/\tau\right)}{\sum_{j=1}^{HW} \exp\left(\text{sim}(f_i^{\text{pfa}}, f_j^{\text{fm}})/\tau\right)}, \tag{11}$$

where $\text{sim}(a, b) = \frac{a^\top b}{\|a\|\|b\|}$, and $\tau$ is the temperature parameter. $f_i^{\text{pfa}}$ and $f_i^{\text{fm}}$ represent the feature vectors at spatial location $i$ (flattened from the 2D grid) extracted from the adapted VQ features and the FM features, respectively, both with dimensionality $C'$. The indices $i \in \{1, \ldots, H'W'\}$ correspond to all patch locations on the $H' \times W'$ spatial grid. The patch-wise formulation enables localized semantic supervision, allowing the model to align not just global representations but also fine-grained spatial semantics. By minimizing $\mathcal{L}_{\text{align}}$, the discretized features are encouraged to retain rich and spatially semantic information that are consistent with the external FM prior, effectively mitigating the loss of detail and semantic drift introduced during quantization. In practice, we adopt DINOv2 [42] as the foundation model to provide semantic supervision.

## 3.6 Total Optimization Objective

Drawing upon the loss formulations in equations 10 and 11, the overall optimization objective of our framework is defined as follows:

$$\mathcal{L} = \mathcal{L}_{\text{db}} + \lambda_a \mathcal{L}_{\text{align}}, \tag{12}$$

where $\lambda_a$ is a hyperparameter that balances the two loss terms. A decrease in $\mathcal{L}_{\text{db}}$ indicates that the model is learning more effective representations for reconstructing images and segmenting relevant lesion regions. Furthermore, the optimization of $\mathcal{L}_{\text{align}}$ suggests that stronger consistency is achieved between quantized features and external foundation model priors, thereby mitigating detail loss and semantic shift introduced during the quantization process. Detailed discussions on the hyperparameter can be found in the ablation study part 4.5.

# 4 Experiments

## 4.1 Datasets.

**Lung Cancer (LC) Dataset.** We collect a multi-center dataset including 828 chest CT scans of central-type lung carcinoma, which reveals the inherent challenges associated with detecting and analyzing such cases, presenting subtle anomalies within the imaging data. There exists one segmentation target per volume, and the dominant lesion is annotated precisely for each case.

**ACDC dataset.** This dataset [43] is a cardiac MRI collection comprising 100 short-axis cine-MRI scans, acquired using both 3T and 1.5T scanners.

Following previous studies, we applied the 70–10–20 split ratio for training, validation, and testing on the LC dataset. To ensure a fair comparison, all experiments were conducted on 2D slices.

## 4.2 Implementation Details.

Our model is implemented using PyTorch 2.0.1 with CUDA 11.8 and MONAI 1.3.0. All 2D slices are cropped to $128 \times 128$ and used as input, with a batch size of 4 per GPU. The training process runs for 100 epochs, employing the cross-entropy loss and an SGD optimizer with a polynomial learning rate scheduler (initial learning rate of $1 \times 10^{-4}$ and decay of $3 \times 10^{-5}$). As shown in Fig. 2, several data augmentation strategies are applied, including random rotation, color jittering, Gaussian noise, and

Table 1: Quantitative comparison on the LC dataset with two labeled ratio settings (5%, 10%) using four metrics: Dice and Jaccard (↑), HD95 and ASD (↓). Best results are in **bold**, second best are underlined.

| Method | 5% Labeled | | | | 10% Labeled | | | |
|---|---|---|---|---|---|---|---|---|
| | Dice↑ | Jaccard↑ | HD95↓ | ASD↓ | Dice↑ | Jaccard↑ | HD95↓ | ASD↓ |
| UNet-F [47] | 0.8345 | 0.7386 | 6.9634 | 2.2913 | 0.8345 | 0.7386 | 6.9634 | 2.2913 |
| UNet-S [47] | 0.4343 | 0.3118 | 26.0498 | 12.6188 | 0.6490 | 0.5175 | 21.4063 | 7.3382 |
| nnUNet-F [54] | 0.8259 | 0.7236 | 4.2533 | 1.4216 | 0.8259 | 0.7236 | 4.2533 | 1.4216 |
| nnUNet-S [54] | 0.4590 | 0.3438 | 13.2746 | 8.8636 | 0.6538 | 0.5194 | 25.2100 | 8.9332 |
| UA-MT [16] | 0.6029 | 0.4647 | 48.6681 | 24.6020 | 0.7222 | 0.5989 | 11.6724 | 5.4939 |
| MCNet [48] | 0.6378 | 0.4970 | 15.2759 | 4.9231 | 0.7555 | 0.6414 | 16.1903 | 9.9647 |
| SSNet [49] | 0.6328 | 0.4886 | 25.1005 | 9.3180 | 0.7480 | 0.6278 | 14.9581 | 7.3399 |
| BCP [50] | 0.6243 | 0.4854 | 26.9303 | 10.4789 | 0.7252 | 0.5994 | 18.9768 | 6.5105 |
| ARCO [51] | 0.6162 | 0.4778 | 36.2256 | 14.6243 | 0.7246 | 0.5945 | 14.4803 | 4.3660 |
| ABD [52] | 0.6414 | 0.5024 | 12.5608 | 5.9661 | 0.7468 | 0.6244 | 12.6570 | 6.7437 |
| Unimatch [53] | 0.6493 | 0.5071 | 17.8700 | 5.4526 | 0.7511 | 0.6333 | 17.0178 | 5.7388 |
| Ours | **0.6643** | **0.5257** | **12.2525** | **4.2276** | **0.7852** | **0.6731** | **11.6179** | **4.2094** |

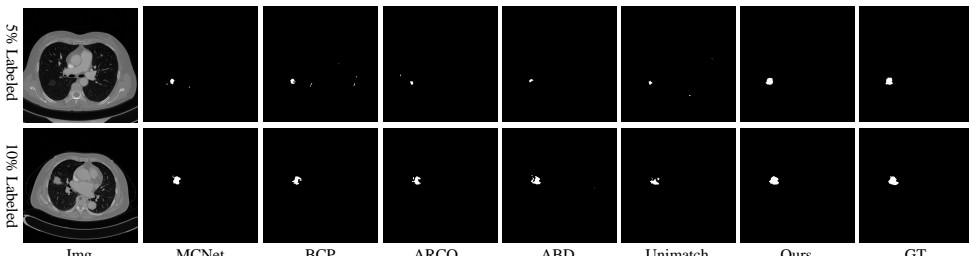

Figure 5: Visual results on LC with 5% and 10% labeled data show that VQ-Seg consistently yields more accurate predictions of anatomical structures and boundaries than all other compared methods.

blurring. To enable gradient backpropagation through the non-differentiable operations in our model, we employ the Straight-Through Estimator (STE) technique [44], which approximates gradients during the backward pass. In our main experiments, we set the VQ codebook size to $K = 16{,}384$ and further conduct corresponding ablation studies (see Sec. 4.5) to evaluate its influence. Moreover, the codebook mapping mechanism [45, 46] is incorporated to accelerate the learning dynamics of the codebook. An Exponential Moving Average (EMA) strategy is applied to update the teacher network, where the parameters of the teacher encoder and decoder are updated from the student's parameters as follows: $\theta_t \leftarrow \alpha \cdot \theta_t + (1 - \alpha) \cdot \theta_s$, where $\theta_t$ and $\theta_s$ represent the parameters of the teacher and student networks, respectively, and $\alpha = 0.99$ denotes the EMA decay rate. All experiments are conducted on a cloud computing platform equipped with four NVIDIA GeForce RTX 4090 GPUs.

## 4.3 Evaluation and Metrics.

We compare our method with other state-of-the-art (SOTA) approaches, including UNet [47], UA-MT [16], MCNet [48], SSNet [49], BCP [50], ARCO [51], ABD [52], and Unimatch [53]. Note that methods labeled with "F" denote fully supervised models trained using all labeled data, while those labeled with "S" indicate semi-supervised models trained with limited annotations. To ensure a fair comparison, our VQ-Seg model adopts the same encoder and decoder architecture as Unimatch [53]. The main difference lies in the introduction of a quantization and alignment process after the encoder output. Segmentation performance is evaluated using four common metrics: Dice and Jaccard for region overlap, and HD95 and ASD for boundary accuracy.

Table 2: A comprehensive ablation study evaluating the contributions of the Quantized Perturbation Module (QPM), the dual-branch architecture with shared Post-VQ space (DB), and the Post-VQ Feature Adapter (PFA) on segmentation performance using the LC dataset (10% labeled).

| Base | QPM | DB | PFA | Dice↑ | Jaccard↑ | HD95↓ | ASD↓ |
|------|-----|----|-----|-------|----------|-------|------|
| ✓ | | | | 0.7443 | 0.6238 | 14.2153 | 5.2301 |
| ✓ | ✓ | | | 0.7701 | 0.6559 | 13.0246 | 4.9378 |
| ✓ | ✓ | ✓ | | 0.7784 | 0.6620 | 12.4728 | 4.6013 |
| ✓ | ✓ | | ✓ | 0.7761 | 0.6597 | 12.7381 | 4.7005 |
| ✓ | ✓ | ✓ | ✓ | **0.7852** | **0.6731** | **11.6179** | **4.2094** |

Table 3: Impact of hyperparameters on metrics.

| Param. | Value | Dice↑ | Jaccard↑ | HD95↓ | ASD↓ |
|--------|-------|-------|----------|-------|------|
| $\epsilon$ | 0.3 | 0.7741 | 0.6612 | 12.3821 | 4.2983 |
| | 0.5 | 0.7803 | 0.6685 | 11.9042 | 4.2150 |
| | 0.7 | **0.7852** | **0.6731** | **11.6179** | **4.2094** |
| | 0.9 | 0.7418 | 0.6142 | 14.8054 | 5.1328 |
| $\lambda_a$ | 1 | 0.7720 | 0.6591 | **11.5080** | 4.2672 |
| | 5 | **0.7852** | **0.6731** | 11.6179 | 4.2094 |
| | 10 | 0.7765 | 0.6640 | 11.9235 | **4.1926** |
| $\lambda_u$ | 1 | **0.7852** | **0.6731** | 11.6179 | 4.2094 |
| | 5 | 0.7670 | 0.6553 | 11.7421 | 4.1652 |
| | 10 | 0.7843 | 0.6724 | 11.8852 | **4.1013** |

## 4.4 Comparison Results

**Quantitative Comparisons.** Tables 1 and 7 summarize the quantitative comparisons for each method on the LC and ACDC datasets across two labeled ratio settings (5% and 10%). On the LC dataset, our method demonstrates superior performance. At 5% labeled data, VQ-Seg achieves the highest Dice (0.6643) and Jaccard (0.5257), outperforming the second-best Unimatch (Dice: 0.6493, Jaccard: 0.5071) by 1.5% and 1.86%, respectively. It also yields the best HD95 (12.2525) and ASD (4.2276), improving upon the second-best ABD (HD95: 12.5608) and MCNet (ASD: 4.9231) by 0.3083 and 0.6955. With 10% labeled data, our method maintains the lead, achieving the highest Dice (0.7852) and Jaccard (0.6731), surpassing the second-best MCNet (Dice: 0.7555, Jaccard: 0.6414) by 2.97% and 3.17%. It also yields the best HD95 (11.6179) and ASD (4.2094), improving upon the second-best UA-MT (HD95: 11.6724) and ARCO (ASD: 4.3660) by 0.0545 and 0.1566. Results on the ACDC dataset (see Appendix C and Table 7) exhibit a similar trend, confirming the robustness and generalizability of our approach across diverse datasets.

**Visual Comparisons.** As depicted in Fig. 5, our VQ-Seg effectively identifies the cancerous areas with high precision. Segmentation outcomes exhibit improved consistency and clearer boundary delineation when compared to other state-of-the-art techniques. Notably, our method better preserves the structural integrity of cancer regions. These visual advantages further demonstrate the superior performance of our model across various cases. For a more comprehensive understanding, please refer to Appendix D for the statistical visualization analysis and Appendix E for the t-SNE visualization of the codebook update dynamics.

## 4.5 Ablation Studies

**Module Analysis.** As shown in Table 2, we conduct a step-by-step ablation to evaluate the contribution of each proposed component. The baseline model is a VQ-embedded Unimatch with a student-teacher framework. Based on this, incorporating the Quantized Perturbation Module (QPM) leads to a notable improvement in Dice from 0.7443 to 0.7701. Adding the dual-branch (DB) architecture further enhances performance to 0.7784, while using the Post-VQ Feature Adapter (PFA) alone yields a Dice of 0.7761. When all three modules are combined, the full model achieves the best performance across all metrics, including a Dice of 0.7852, Jaccard of 0.6731, HD95 of 11.6179, and ASD of 4.2094, confirming the complementary benefits of the proposed components.

**Hyper-parameter Experiment.** Table 3 reports the impact of key hyperparameters. For the perturbation strength $\epsilon$, performance improves as $\epsilon$ increases to 0.7, achieving the best Dice score of 0.7852. Regarding the loss weights, setting $\lambda_a = 5$ and $\lambda_u = 1$ leads to the best overall results.

**Effect of Foundation Models.** We choose DINOv2 as our backbone because it has demonstrated remarkable generalization across diverse downstream tasks [55, 56]; despite being trained on natural images, it transfers strongly to medical domains and has been widely used in medical segmentation and registration. To further validate this design choice, we conducted an ablation replacing DINOv2 with alternative foundation models, including those pretrained on medical data. CLIP [57] and BiomedCLIP [58] are vision–language models trained on large-scale image–text pairs (natural and medical, respectively), MAE [59] is trained via masked autoencoding, and Rad-DINO [60] adopts a DINO-style architecture on radiology images. As summarized in Table 4, DINOv2 consistently outperforms all alternatives under both 5% and 10% labeled regimes, including models specialized

Table 4: Ablation on foundation models.

| Foundation Model | Dice (5%) | Dice (10%) |
|---|---|---|
| CLIP [57] | 0.6421 | 0.7483 |
| BiomedCLIP [58] | 0.6507 | 0.7629 |
| MAE [59] | 0.6386 | 0.7541 |
| Rad-DINO [60] | 0.6535 | 0.7793 |
| **DINOv2** [42] | **0.6643** | **0.7852** |

Table 5: Ablation on the codebook size.

| Codebook Size | Dice (5%) | Dice (10%) | Uti. (%) |
|---|---|---|---|
| 1024 | 0.6531 | 0.7608 | **100** |
| 2048 | 0.6582 | 0.7748 | **100** |
| 4096 | 0.6627 | 0.7775 | 99 |
| 16384 | **0.6643** | **0.7852** | 98 |
| 32768 | 0.6595 | 0.7764 | 95 |
| 65536 | 0.6415 | 0.7638 | 92 |

Table 6: Comparison of performance under different labeled data ratios.

| Method | 5% | 10% | 20% | 50% | 100% |
|---|---|---|---|---|---|
| UNet-S [47] | 0.4343 | 0.6490 | 0.7205 | 0.7880 | 0.8345 |
| MCNet [48] | 0.6378 | 0.7555 | 0.7812 | 0.8203 | 0.8751 |
| ABD [52] | 0.6414 | 0.7468 | 0.7780 | 0.8235 | 0.8824 |
| Unimatch [53] | 0.6493 | 0.7511 | 0.7855 | 0.8279 | 0.8871 |
| **VQ-Seg (Ours)** | **0.6643** | **0.7852** | **0.8100** | **0.8507** | **0.9102** |

for medical domains, empirically supporting DINOv2 as a robust and effective semantic prior for semi-supervised medical image segmentation.

**Effect of Codebook Size.** We further investigate the impact of codebook size on model performance, as shown in Table 5. The codebook size determines the granularity of the discrete latent space: a smaller codebook provides limited representational capacity, while an excessively large one may lead to code redundancy and unstable optimization. Our results show that the Dice score steadily increases as the codebook size grows from 1,024 to 16,384, indicating that a moderately large codebook enables richer and more discriminative representations. However, further enlargement (e.g., 32,768 or 65,536) slightly degrades performance due to decreased code utilization and overfitting. Here, "Uti." denotes codebook utilization, which measures the proportion of code vectors actively used during training.

**Ablation on Labeled Ratio Settings.** As shown in Table 6, we evaluate VQ-Seg under different labeled ratios to examine its scalability in semi-supervised learning. The results show a consistent performance gain with more labeled data, demonstrating the model's strong ability to exploit supervision. Remarkably, VQ-Seg achieves significant improvements in low-label regimes, indicating that the discrete representation learned via vector quantization provides effective regularization and semantic consistency. Even with increasing supervision, the performance gain remains stable, highlighting VQ-Seg's robustness and scalability across varying annotation levels.

## 5 Conclusion and Limitations

We present VQ-Seg, a novel semi-supervised medical image segmentation framework. VQ-Seg introduces a Quantized Perturbation Module (QPM) that performs controlled perturbations in the vector-quantized (VQ) feature space, enhancing the robustness of representation learning. In addition, a dual-branch architecture with a Post-VQ Feature Adapter (PFA) is designed to refine the quantized features and integrate high-level semantic information. Extensive experiments on the Lung Cancer (LC) and ACDC datasets demonstrate that VQ-Seg achieves state-of-the-art performance, substantially improving segmentation accuracy under limited supervision.

However, the current perturbation operates solely in the discrete VQ space, making it difficult to extend to continuous feature representations commonly used in existing semi-supervised frameworks. Moreover, while the adoption of a foundation model introduces richer semantic priors, it also brings additional computational overhead. Future work will focus on developing controllable perturbation mechanisms in continuous spaces and exploring more efficient foundation model integration.

## Acknowledgments and Disclosure of Funding

This work is supported by the Guangdong Science and Technology Department (2024ZDZX2004) and the Guangzhou Industrial Information and Intelligent Key Laboratory Project (No. 2024A03J0628).

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

# A  KL Divergence Approximation under Dropout Perturbation

To support the theoretical analysis in Section 3.2, we derive an approximation of the KL divergence between a prior distribution $P(h)$ and a perturbed distribution $Q(h)$ induced by applying dropout to intermediate feature. This divergence is used to quantify the perturbation radius caused by dropout in the feature space.

## A.1  Dropout-Induced Feature Distribution as a Mixture

Consider a simplified setting where a feature activation $h$ follows a Gaussian prior:

$$P(h) = \mathcal{N}(0, \sigma_0^2). \tag{13}$$

Under dropout with rate $p$, the feature is zeroed out with probability $p$, or retained with probability $1 - p$. This results in the perturbed distribution:

$$Q(h) = (1 - p) \cdot \mathcal{N}(0, \sigma_0^2) + p \cdot \delta(h), \tag{14}$$

where $\delta(h)$ is the Dirac delta function centered at zero. This mixture captures the effect of randomly dropping activations.

## A.2  Intractability of Exact KL Divergence

The KL divergence between $P(h)$ and $Q(h)$ is:

$$D_{\mathrm{KL}}(P\|Q) = \int_{-\infty}^{\infty} P(h) \log\left(\frac{P(h)}{Q(h)}\right) dh. \tag{15}$$

Substituting the forms of $P(h)$ and $Q(h)$, we obtain:

$$D_{\mathrm{KL}}(P\|Q) = \int_{-\infty}^{\infty} \mathcal{N}(h; 0, \sigma_0^2) \log\left(\frac{\mathcal{N}(h; 0, \sigma_0^2)}{(1-p)\mathcal{N}(h; 0, \sigma_0^2) + p\delta(h)}\right) dh. \tag{16}$$

Due to the singular nature of $\delta(h)$ at $h = 0$, this expression is intractable in closed form. To make progress, we adopt a moment-matching approximation.

## A.3  Moment-Matching Approximation

We approximate $Q(h)$ with a Gaussian distribution $Q_{\mathrm{approx}}(h)$ that matches the first and second moments of the dropout-perturbed activations. Let $h' \sim Q(h)$ denote the post-dropout feature. Then:

$$\mathbb{E}[h'] = 0, \quad \mathrm{Var}(h') = \sigma_0^2(1 - p). \tag{17}$$

Therefore, we define:

$$Q_{\mathrm{approx}}(h) = \mathcal{N}(0, \sigma_0^2(1 - p)). \tag{18}$$

The KL divergence between the original and approximated feature distributions becomes:

$$D_{\mathrm{KL}}(P\|Q_{\mathrm{approx}}) = \frac{1}{2}\left(\frac{\sigma_0^2}{\sigma_0^2(1 - p)} - 1 + \log\left(\frac{\sigma_0^2(1 - p)}{\sigma_0^2}\right)\right). \tag{19}$$

Simplifying gives:

$$D_{\mathrm{KL}}(P\|Q_{\mathrm{approx}}) = \frac{1}{2}\left(\frac{1}{1 - p} - 1 + \log(1 - p)\right) = \frac{1}{2}\left(\frac{p}{1 - p} + \log(1 - p)\right). \tag{20}$$

## A.4  Interpretation and Connection to Perturbation Radius

This result provides a clean, analytical expression for the perturbation radius induced by dropout, interpreted as a KL divergence. Notably, as $p \to 1$, the divergence grows rapidly and even diverges (.$e.$, becomes unbounded), indicating severe deviation from the original distribution. This supports our theoretical claim: Dropout induces increasingly unstable perturbations when the dropout rate is high, as observed empirically (see Fig. 1), potentially leading to over-regularization, distorted representations, and degraded model performance. This motivates our Quantized Perturbation Module (QPM), which constrains perturbations to a structured, discrete space and avoids such instability.

# B  Proof of Numerical Stability of QPM

In this appendix, we prove that the perturbed distribution $Q(c_j \mid \epsilon)$ used in the Quantized Perturbation Module (QPM) is always well-defined and bounded, thereby ensuring the numerical stability of the associated KL divergence, even in the extreme case of $\epsilon = 1$.

## B.1  Definition of $Q(c_j \mid \epsilon)$

Let the prior distribution over codewords be uniform:

$$P(c_i) = \frac{1}{K}, \quad \forall i \in \{1, \dots, K\}. \tag{21}$$

The perturbation mechanism $\pi(j \mid i)$ is defined as:

$$\pi(j \mid i) = \begin{cases} 1 - \epsilon, & \text{if } j = i \\ \frac{\epsilon \exp(-d(c_i, c_j))}{Z_i}, & \text{if } j \neq i \end{cases}, \quad \text{where } Z_i = \sum_{k \neq i} \exp(-d(c_i, c_k)). \tag{22}$$

Then, the overall perturbed distribution is:

$$Q(c_j \mid \epsilon) = \sum_{i=1}^{K} P(c_i)\pi(j \mid i) = \frac{1}{K} \sum_{i=1}^{K} \pi(j \mid i). \tag{23}$$

## B.2  Basic Properties of $Q(c_j \mid \epsilon)$

**(1) Non-negativity and Positivity:**  Since $\pi(j \mid i) \geq 0$ for all $i, j$, it follows that $Q(c_j \mid \epsilon) \geq 0$. Furthermore, under mild assumptions (e.g., finite distances and $\epsilon > 0$), we have $\pi(j \mid i) > 0$ for some $i$, so $Q(c_j \mid \epsilon) > 0$ for all $j$.

**(2) Normalization:**  We verify that $Q$ is a valid probability distribution:

$$\sum_{j=1}^{K} Q(c_j \mid \epsilon) = \sum_{j=1}^{K} \sum_{i=1}^{K} \frac{1}{K} \pi(j \mid i) = \frac{1}{K} \sum_{i=1}^{K} \sum_{j=1}^{K} \pi(j \mid i) = \frac{1}{K} \sum_{i=1}^{K} 1 = 1. \tag{24}$$

## B.3  KL Divergence Between $P$ and $Q$

The KL divergence between $P$ and $Q$ is defined as:

$$D_{\text{KL}}(P \parallel Q) = \sum_{j=1}^{K} P(c_j) \log\left(\frac{P(c_j)}{Q(c_j \mid \epsilon)}\right) = \frac{1}{K} \sum_{j=1}^{K} \log\left(\frac{1/K}{Q(c_j \mid \epsilon)}\right) \tag{25}$$

$$= -\frac{1}{K} \sum_{j=1}^{K} \log\left(K Q(c_j \mid \epsilon)\right). \tag{26}$$

This expression is numerically stable as long as $Q(c_j \mid \epsilon) > 0$ and bounded away from 0, which we prove next for the extreme case $\epsilon = 1$.

## B.4  QPM Perturbation Distribution at $\epsilon = 1$

Recall that when $\epsilon = 1$, the transition distribution becomes:

$$\pi(j \mid i) = \begin{cases} 0, & \text{if } j = i \\ \frac{\exp(-d(c_i, c_j))}{Z_i}, & \text{if } j \neq i \end{cases}, \quad \text{where } Z_i = \sum_{k \neq i} \exp(-d(c_i, c_k)). \tag{27}$$

Thus, the resulting perturbed distribution is given by:

$$Q(c_j \mid \epsilon = 1) = \sum_{i=1}^{K} P(c_i)\pi(j \mid i) = \frac{1}{K} \sum_{i \neq j} \frac{\exp(-d(c_i, c_j))}{\sum_{k \neq i} \exp(-d(c_i, c_k))}. \tag{28}$$

**B.5    Lower Bound of $Q(c_j \mid \epsilon = 1)$**

Assume the codebook $\mathcal{C} = \{c_1, ..., c_K\}$ is fixed and that the pairwise distance is bounded: $0 < D_{\min} \leq d(c_i, c_j) \leq D_{\max} < \infty$ for all $i \neq j$. Then for any $i \neq j$, the transition term is lower bounded:

$$\frac{\exp(-d(c_i, c_j))}{\sum_{k \neq i} \exp(-d(c_i, c_k))} \geq \frac{\exp(-D_{\max})}{(K-1)\exp(-D_{\min})} = \frac{\exp(D_{\min} - D_{\max})}{K-1}. \tag{29}$$

Hence,

$$Q(c_j \mid \epsilon = 1) = \frac{1}{K} \sum_{i \neq j} \frac{\exp(-d(c_i, c_j))}{\sum_{k \neq i} \exp(-d(c_i, c_k))} \tag{30}$$

$$\geq \frac{1}{K}(K-1) \cdot \frac{\exp(D_{\min} - D_{\max})}{K-1} = \frac{\exp(D_{\min} - D_{\max})}{K} > 0. \tag{31}$$

**B.6    Upper Bound of $Q(c_j \mid \epsilon = 1)$**

Likewise, for any $i \neq j$:

$$\frac{\exp(-d(c_i, c_j))}{\sum_{k \neq i} \exp(-d(c_i, c_k))} \leq \frac{\exp(-D_{\min})}{(K-1)\exp(-D_{\max})} = \frac{\exp(D_{\max} - D_{\min})}{K-1}. \tag{32}$$

Thus,

$$Q(c_j \mid \epsilon = 1) \leq \frac{1}{K}(K-1) \cdot \frac{\exp(D_{\max} - D_{\min})}{K-1} = \frac{\exp(D_{\max} - D_{\min})}{K}. \tag{33}$$

**B.7    Implication for KL Divergence**

Since $Q(c_j \mid \epsilon = 1)$ is strictly positive and bounded above by 1, the term $\log(KQ(c_j))$ in the KL divergence remains finite:

$$D_{\mathrm{KL}}(P \,\|\, Q) = -\frac{1}{K} \sum_{j=1}^{K} \log(KQ(c_j)) < \infty. \tag{34}$$

Therefore, the QPM perturbation strategy ensures numerical stability for all valid $\epsilon \in [0, 1]$.

## C    Performance Analysis on the ACDC Dataset

Table 7: Quantitative comparison on the ACDC dataset with two labeled ratio settings (5%, 10%) using Dice and Jaccard ($\uparrow$). Best results are in **bold**.

| Method | 5% Labeled | | 10% Labeled | |
|---|---|---|---|---|
| | Dice$\uparrow$ | Jaccard$\uparrow$ | Dice$\uparrow$ | Jaccard$\uparrow$ |
| UNet-F [47] | 0.9130 | 0.8427 | 0.9130 | 0.8427 |
| UNet-S [47] | 0.4674 | 0.3698 | 0.7952 | 0.6882 |
| nnUNet-F [54] | 0.9185 | 0.8491 | 0.9185 | 0.8491 |
| nnUNet-S [54] | 0.4892 | 0.3876 | 0.8113 | 0.7040 |
| UA-MT [16] | 0.6123 | 0.5324 | 0.8423 | 0.7382 |
| MCNet [48] | 0.6485 | 0.5338 | 0.8621 | 0.7701 |
| SSNet [49] | 0.6542 | 0.5568 | 0.8689 | 0.7753 |
| BCP [50] | 0.8621 | 0.7846 | 0.8827 | 0.8032 |
| ARCO [51] | 0.8879 | 0.8021 | 0.9026 | 0.8252 |
| ABD [52] | 0.8874 | 0.7924 | 0.8992 | 0.8213 |
| Unimatch [53] | 0.8915 | 0.7983 | 0.8978 | 0.8290 |
| Ours | **0.9057** | **0.8173** | **0.9103** | **0.8327** |

Table 7 compares our method with leading baselines on the ACDC dataset under 5% and 10% labeled data settings. Our model consistently outperforms others in both Dice and Jaccard metrics.

With 5% labeled data, our method achieves a Dice of 0.9057, exceeding Unimatch (0.8915), ABD (0.8874), and ARCO (0.8879), and approaching the fully supervised nnUNet-F (0.9185). This highlights its strong representation ability under limited supervision.

At 10% labeled data, the advantage becomes more evident, reaching the best Dice (0.9103) and Jaccard (0.8327), surpassing ABD (0.8992) and ARCO (0.9026). These consistent gains demonstrate the robustness and scalability of our approach as more labeled data are available.

## D  Statistical Analysis

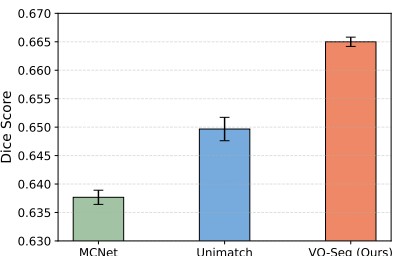

Figure 6: Comparison under 5% labeled LC dataset.

To evaluate the statistical reliability of our method, we ran the competing models MCNet [48], Unimatch [53], and our VQ-Seg ten times under the 5% labeled LC dataset using different random seeds. The averaged Dice scores together with their standard deviations are visualized in Fig. 6, where the *error bars* denote the standard deviation across repeated runs, reflecting the robustness and stability of each method. The noticeably smaller error bar of VQ-Seg indicates lower performance variance, demonstrating its stronger training stability across random seeds.

To confirm the statistical significance of the observed improvements, we performed paired two-tailed $t$-tests between VQ-Seg and each baseline method over the ten independent trials. The results show that the differences are statistically significant with $p < 0.05$.

## E  Codebook Evolution

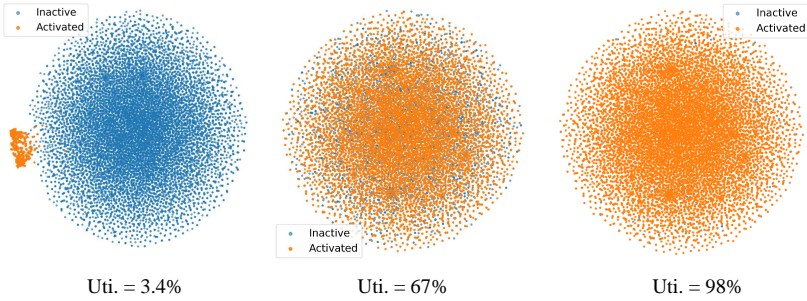

Figure 7: T-SNE visualization of codebook evolution.

We visualize in Fig. 7 the t-SNE projections of all codebook vectors across three training stages. Each point represents a codeword, where orange and blue indicate *activated* and *inactive* entries, respectively. Initially (left), only a few codewords (3.4%) are activated, showing a compact cluster and limited diversity. As training proceeds (middle), activation increases to 67%, and the distribution becomes more uniform. By convergence (right), nearly all codewords (98%) are active and evenly dispersed, forming a stable and well-structured embedding space.

These results demonstrate that our quantization strategy progressively enhances codebook utilization and representation diversity.

