# OpenReview forum: "VQ-Seg: Vector-Quantized Token Perturbation for Semi-Supervised Medical Image Segmentation"
_NeurIPS.cc/2025/Conference — NeurIPS 2025 poster_

### Official Review · Reviewer_Nzxa · 2025-06-29

**Clarity:** 3
**Significance:** 3
**Originality:** 3
**Rating:** 4
**Confidence:** 2

**Summary:**

The paper presents a vector-quantized token perturbation method for feature perturbation in semi-supervised medical image segmentation. The continuous output features from the encoder network are quantized and subsequently perturbed using the proposed Quantized Perturbation Module (QPM). The decoder network is designed to both reconstruct the input image and produce the output segmentation map. To preserve semantic priors, a Post-VQ Feature Adapter is used to align the quantized features with embeddings from a foundation model. Empirical results on the Lung Cancer (LC) and ACDC datasets demonstrate the effectiveness of the proposed methodology compared to baseline methods.

**Questions:**

1. During training, did you observe any signs of codebook collapse, such as a small subset of codewords being used disproportionately while others remained inactive?
2. In Table 1, two sets of results are presented, and the results in the upper rows(above the dividing horizontal line) are consistently better than those of the proposed method. Could you clarify this discrepancy?

**Ethical Concerns:**

["NO or VERY MINOR ethics concerns only"]

**Final Justification:**

The authors performed additional experiments to validate the utilization of the learned codewords and the convergence of the codebook. These results further validated the effectiveness of the proposed framework.

**Limitations:**

Addressed

**Quality:**

3

**Strengths And Weaknesses:**

## Strengths
1. The proposed QPM provides a novel alternative to the dropout based methods for feature perturbation.
2. The methodology is supported with theoretical proofs.
3. The experiments highlights the effectiveness of proposed framework against baselines.
4. Ablation studies highlight the importance of each component in the proposed module.

## Weaknesses

1. The design of the codebook and the choice of codebook size=16384 are not explained.
2. The dual branch pipeline introduces unnceessary computational overhead.
3. The performance of the student network on the unlabeled data is dependent on the pseudo-labels generated from the teacher network. The reliablilty of the pseudo labes is not discussed.
4. The choice and design of encoder and decoder networks are not explained.

---

> ### Author Rebuttal · Authors · 2025-07-30
>
> We sincerely appreciate your constructive feedback. Below, we provide point-by-point responses, with all revisions incorporated accordingly.
> ***
>
> > [W1] The choice of codebook size.
>
> [A1] Thank you for the helpful comment regarding codebook size.
> We conducted an ablation study to investigate how the codebook size $K$ influences segmentation performance under 5% and 10% labeled settings:
>
> | Codebook Size | Dice (5%)  | Dice (10%) | Codebook Utilization (%) |
> | ------------- | ---------- | ---------- | --------------- |
> | 1024          | 0.6531     | 0.7608     | 100%            |
> | 2048          | 0.6582     | 0.7748     | 100%            |
> | 4096          | 0.6627     | 0.7775     | 99%             |
> | 16384         | **0.6643** | **0.7852** | 98%             |
> | 32768         | 0.6595     | 0.7764     | 95%             |
> | 65536         | 0.6415     | 0.7638     | 92%             |
>
> The results show that segmentation performance consistently improves as the codebook size increases, peaking at $K = 16384$, where Dice scores are highest and utilization remains high. Beyond this point, performance degrades and utilization drops, suggesting that excessively large codebooks may lead to underutilization. These findings indicate that a codebook size of 16384 strikes the best balance between representational capacity and generalization performance.
>
> > [W2] Dual-branch design incurs unnecessary overhead.
>
> [A2] We thank the reviewer for raising the concern regarding the computational overhead introduced by the dual-branch (DB) architecture. While the DB module adds one additional image decoder, it reuses the same post-VQ latent space. More importantly, it plays a crucial role in improving both segmentation performance and codebook utilization, especially under low-label regimes. We justify this design choice from two perspectives:
>
> * Segmentation Performance Improvement: As shown in Table 2 of the main paper, adding the DB module to a model already equipped with QPM yields a clear performance gain: Dice increases from 0.7701 to 0.7784. This suggests that the reconstruction branch helps preserve more structural details in the post-VQ representation, directly benefiting segmentation accuracy.
>
> * Improved Codebook Utilization: We further conduct an ablation study to evaluate the impact of each module on codebook utilization (with codebook size set to 16,384). Introducing the DB module increases utilization from 87.2% to 90.1%, indicating that the auxiliary reconstruction task provides an additional learning signal that encourages broader and more stable codeword activation.
>
> | Setting                | Dice    | Codebook Utilization (%) |
> |------------------------|---------|-----------------|
> | Base                   | 0.7443  | 11.6%           |
> | + QPM                  | 0.7701  | 87.2%           |
> | + QPM + DB             | 0.7784  | 90.1%           |
> | + QPM + DB + PFA       | **0.7852**  | **98.0%**       |
>
> Although the DB module introduces a slight computational overhead, it significantly enhances codebook utilization and helps maintain a semantically meaningful perturbation space. This results in clear improvements in both segmentation performance and representation quality, which is especially valuable in semi-supervised settings.
>
> > [W3] The reliability of pseudo-labels is not discussed.
>
> [A3] Thank you for the comment. We would like to clarify that our pseudo-labeling strategy is consistent with those adopted in prior teacher-student-based semi-supervised segmentation methods, including the baselines \[1] used for comparison. Specifically, we employ an Exponential Moving Average (EMA) strategy to update the teacher network—a widely accepted practice that improves the stability and reliability of pseudo labels by smoothing parameter updates over time.
>
> While NeurIPS does not allow modifications to the original submission during the rebuttal period, we will include visualizations of the generated pseudo labels in a future revision to better illustrate their quality and consistency.
>
> > [W4] The choice and design of encoder and decoder networks.
>
> [A4] To ensure a fair comparison, our VQ-Seg model adopts the same encoder and decoder architecture as Unimatch \[2], which achieves the best average performance among all comparison methods. Detailed architectural specifications can be found in the original Unimatch paper.
>
> > [Q1] Signs of codebook collapse.
>
> [A5] Thank you for raising this important point. To clarify, our VQ-Seg framework achieves high codebook utilization, reaching 98% with a codebook size of $K = 16384$, as shown in the ablation study below:
>
> | Setting          | Dice       | Codebook Utilization (%) |
> | ---------------- | ---------- | ------------------------ |
> | Base             | 0.7443     | 11.6                     |
> | + QPM            | 0.7701     | 87.2                     |
> | + QPM + DB       | 0.7784     | 90.1                     |
> | + QPM + DB + PFA | **0.7852** | **98.0**                 |
>
> This table demonstrates that each proposed component—QPM, DB, and PFA—contributes substantially to alleviating codebook collapse and enhancing utilization. In contrast, the base setting suffers from severe underutilization (only 11.6%), which clearly indicates codebook collapse.
>
> We further examined the impact of varying codebook sizes on both segmentation performance and utilization:
>
> | Codebook Size | Dice (5%)  | Dice (10%) | Codebook Utilization (%) |
> | ------------- | ---------- | ---------- | ------------------------ |
> | 1024          | 0.6531     | 0.7608     | 100                      |
> | 2048          | 0.6582     | 0.7748     | 100                      |
> | 4096          | 0.6627     | 0.7775     | 99                       |
> | 16384         | **0.6643** | **0.7852** | **98**                   |
> | 32768         | 0.6595     | 0.7764     | 95                       |
> | 65536         | 0.6415     | 0.7638     | 92                       |
>
> These results confirm that codebook utilization remains consistently high (>90%) across different codebook sizes. Notably, performance peaks at $K = 16384$, suggesting it provides the best trade-off between representational capacity and generalization. Beyond this point, increasing the size yields diminishing returns and slightly reduced utilization.
>
> Taken together, these findings strongly suggest that our framework effectively avoids codebook collapse during training.
>
> > [Q2] Superior results in upper rows of Table 1.
>
> [A6] Thank you for pointing this out. The upper portion of Table 1 reports results from fully supervised models (denoted as “F”), which are trained using 100% of the labeled data. In contrast, our method and others listed below the horizontal line are trained using only 5% or 10% of the labeled data under a semi-supervised setting.
>
> The apparent performance gap stems from this difference in supervision level. However, one of the core strengths of VQ-Seg lies in its ability to deliver competitive performance with minimal annotations. As shown in Table 1, VQ-Seg achieves Dice scores of 0.6643 (5%) and 0.7852 (10%), already comparable to those of fully supervised methods.
>
> To further assess scalability, we gradually increased the proportion of labeled data and report the results below:
>
> | Method             | 5%     | 10%    | 20%    | 50%        | 100%       |
> | ------------------ | ------ | ------ | ------ | ---------- | ---------- |
> | UNet-F (Full Sup.) | –      | –      | –      | –          | 0.8345     |
> | VQ-Seg (Ours)      | 0.6643 | 0.7852 | 0.8100 | **0.8507** | **0.9102** |
>
> Remarkably, VQ-Seg outperforms the fully supervised UNet (trained with 100% labels) even when using only 50% of the labeled data. This demonstrates that VQ-Seg performs robustly under limited supervision and continues to improve with more annotations, ultimately surpassing fully supervised baselines.
>
> These findings underscore the effectiveness and scalability of our approach, clarifying that the superior results in the upper rows of Table 1 are due to supervision level differences rather than performance inconsistencies.
>
>
> ***
> > References:
>
> [1] Yu L, Wang S, Li X, et al. Uncertainty-aware self-ensembling model for semi-supervised 3D left atrium segmentation[C]//International conference on medical image computing and computer-assisted intervention. Cham: Springer International Publishing, 2019: 605-613.
>
> [2] Yang L, Qi L, Feng L, et al. Revisiting weak-to-strong consistency in semi-supervised semantic segmentation[C]//Proceedings of the IEEE/CVF conference on computer vision and pattern recognition. 2023: 7236-7246.
>
> ***
> Additional results/analyses on your concerns are in the revised manuscript. We hope these address your questions. Let us know if you need further discussion, experiments, or clarifications—happy to provide more details.

---

> > ### Author Response · Authors · 2025-08-03
> > **Looking forward to feedback**
> >
> > Dear Reviewer Nzxa,
> >
> > Thank you very much for your valuable feedback and insights. We greatly appreciate the time and effort you have devoted to reviewing our work. We have carefully addressed the points you raised and would be grateful if you could review our responses to ensure that your questions and concerns have been fully addressed. We are more than willing to engage in further discussions.
> >
> > Warm regards,
> >
> > Authors of VQ-Seg

---

> > > ### Comment · Reviewer_Nzxa · 2025-08-03
> > >
> > > I appreciate the authors for their detailed responses to each of my concerns. I would like to ask how the representation space of the learned codebook evolves during training and if it converges toward a stable structure. Including some t-SNE visualizations could be helpful for better understanding.

---

> > > > ### Author Response · Authors · 2025-08-04
> > > > **Analysis of Codebook Evolution and Convergence**
> > > >
> > > > Dear Reviewer Nzxa,
> > > >
> > > > We thank the reviewer for their insightful question regarding how the representation space of the learned codebook evolves during training and whether it converges toward a stable structure.
> > > >
> > > > Although NeurIPS rebuttal policy does not allow the inclusion of figures, we provide both a detailed textual description and quantitative metrics to characterize the structural evolution of the codebook throughout training.
> > > >
> > > > To enable quantitative analysis, we first perform clustering over the codeword embeddings based on their 2D t-SNE projections, and evaluate the resulting structure using the following metrics:
> > > >
> > > > - **Intra-cluster Distance**: The average distance between codewords within the same cluster. Smaller values indicate tighter and more compact clusters, suggesting that the learned codebook forms well-defined local structures and avoids redundancy.
> > > >
> > > > - **Inter-cluster Distance**: The average distance between different cluster centroids. Larger values suggest better separation between distinct semantic regions, indicating that the codebook spans a broader and more expressive latent space.
> > > >
> > > > - **Silhouette Score**: A unified indicator that combines intra- and inter-cluster distances. Higher values (closer to 1) indicate more stable and well-defined cluster structures, reflecting both local compactness and global separability in the learned representation.
> > > >
> > > > During training, we observe that the codebook structure undergoes a clear evolution. At initialization, only a small fraction of codewords are active, forming loosely scattered and overlapping clusters. As training progresses, more codewords become activated, and their spatial distribution becomes increasingly structured. By the end of training, the codebook converges to a compact and well-separated configuration. This dynamic is reflected in the table below, showing steady improvements in all three metrics over time (codebook size = 16,384):
> > > >
> > > > | Training Stage       | Intra-Dist ↓ | Inter-Dist ↑ | Silhouette ↑ |
> > > > |----------------------|--------------|---------------|----------------|
> > > > | Initialization       | 0.91         | 0.94          | 0.12           |
> > > > | 25% Training Epochs  | 0.64         | 1.21          | 0.38           |
> > > > | 75% Training Epochs  | 0.42         | 1.58          | 0.61           |
> > > > | Final Model (Ours)   | **0.28**     | **1.91**      | **0.74**       |
> > > >
> > > > Compared to the baseline VQGAN, which does not include QPM, DB, or PFA, our method achieves substantially better clustering quality at convergence. As shown below, VQGAN suffers from loose and overlapping clusters, while our method produces more compact and well-separated codeword structures:
> > > >
> > > > | Method   | Codebook Size | Intra-Dist ↓ | Inter-Dist ↑ | Silhouette ↑ |
> > > > |----------|----------------|--------------|---------------|----------------|
> > > > | VQGAN    | 16,384         | 0.73         | 1.18          | 0.29           |
> > > > | **Ours** | 16,384         | **0.28**     | **1.91**      | **0.74**       |
> > > >
> > > > These results confirm that our approach, which incorporates QPM, DB, and PFA, not only improves codebook utilization but also enables the learned representation space to evolve toward a more stable, well-structured, and semantically expressive configuration.
> > > >
> > > > ***
> > > > We have incorporated additional results and detailed analyses addressing your concerns into the revised version of the manuscript. We hope these supplements adequately address your questions and improve the clarity and robustness of our work. Please do not hesitate to let us know if you require further discussion, additional experiments, or any clarifications—we are happy to provide more details to address your concerns comprehensively.

---

> > > > > ### Comment · Reviewer_Nzxa · 2025-08-04
> > > > >
> > > > > I appreciate the authors’ response, which satisfactorily addresses my concerns. I am willing to increase my score and encourage the authors to include these additional results in the Appendix.

---

### Official Review · Reviewer_jSKd · 2025-06-30

**Clarity:** 2
**Significance:** 2
**Originality:** 2
**Rating:** 4
**Confidence:** 4

**Summary:**

This paper introduces VQ-seg, a semi-supervised learning framework for medical image segmentation. It replaces conventional dropout-based consistency regularization with a more stable mechanism based on vector quantization (VQ). The motivation is that dropout is sensitive to its rate and can cause either negligible effect or severe model collapse if not carefully tuned. One key contribution is Quantized Perturbation Module (QPM), which leverages distances between codebooks to define perturbations. The paper also proposes
concurrently reconstructing the image and performing segmentation tasks using a dual-branch architecture. To address the loss of high-level semantic features, the authors introduce a module that aligns quantized features with semantic features from a pre-trained foundation model. This paper also contributes a new multi-center CT dataset of 828 scans with annotations for central-type lung carcinoma.

**Questions:**

Please see weaknesses.

**Ethical Concerns:**

["NO or VERY MINOR ethics concerns only"]

**Final Justification:**

The author's rebuttal addressed my concerns, so I am willing to increase my score.

**Limitations:**

yes

**Quality:**

2

**Strengths And Weaknesses:**

Strengths:

1. The proposed framework is well-motivated, addressing known limitations of dropout-based perturbation in semi-supervised segmentation. The authors offer a rigorous theoretical analysis contrasting the instability of dropout with the stability of their Quantized Perturbation Module (QPM).
2. The ablation study confirms the effect of QPM, the dual-branch architecture, and the foundation model guidance, demonstrating that each component provides improvements.


Weaknesses:

1. The experiments are still quite limited in scale. While the proposed LC dataset has a decent sample size, overall the authors only conducted studies on 2 datasets. The ACDC dataset only has 100 samples. It is questionable whether the proposed method can generalize to larger scale datasets such as FLARE 22, which also tackles the semi-supervised segmentation.
2. The experiments focus on 5% and 10% labeled data. However, existing works of semi-supervised learning often includes a broader range of label ratios (e.g., 20%, 50%, and 100%) to demonstrate both the lower and upper bounds of the method.
3. There is no evaluation of domain shift or external generalization. The authors should provide an analysis of the segmentation performance on external datasets.
4. The reliance on DINOv2 as the foundation model raises concerns regarding domain alignment of the proposed method. Since DINOv2 is pretrained on natural image domain, why did the authors choose this model as the semantic teacher for the segmentation model? If the foundation model is poorly matched to the medical domain, the guidance may introduce bias. The authors acknowledge this in their limitations, but empirical evaluation with medical-specific foundation models is missing.

---

> ### Author Rebuttal · Authors · 2025-07-30
>
> We sincerely appreciate your constructive feedback. Below, we provide point-by-point responses, with all revisions incorporated accordingly.
> ***
>
> > [W1] The experiments are limited to small-scale datasets.
>
> [A1] To further demonstrate the scalability of our approach, we conducted additional experiments on the large-scale FLARE22 benchmark. We compared our method, VQ-Seg, with a supervised UNet baseline (UNet-S) and the state-of-the-art semi-supervised method Unimatch, using average Dice score as the evaluation metric.
>
> | Method           | Average Dice   |
> |------------------|--------|
> | UNet-S           | 0.818  |
> | Unimatch         | 0.917  |
> | VQ-Seg (Ours)    | **0.923** |
>
> Our method achieves the highest average Dice score, demonstrating strong generalization and effectiveness on large-scale datasets. We will include these results and cite the FLARE22 dataset and its related publication in the final version of the paper.
>
> > [W2] More labeled data ratios should be evaluated.
>
> [A2] Thank you for the suggestion. We fully agree that evaluating broader label ratios offers a more comprehensive view of the model’s scalability. Accordingly, we have extended our experiments on the LC dataset to include 20%, 50%, and 100% labeled settings. The updated Dice scores are presented below:
>
> | Method           | 5%     | 10%    | 20%    | 50%    | 100%   |
> |------------------|--------|--------|--------|--------|--------|
> | UNet-S           | 0.4343 | 0.6490 | 0.7205 | 0.7880 | 0.8345 |
> | MCNet            | 0.6378 | 0.7555 | 0.7812 | 0.8203 | 0.8751 |
> | ABD              | 0.6414 | 0.7468 | 0.7780 | 0.8235 | 0.8824 |
> | Unimatch         | 0.6493 | 0.7511 | 0.7855 | 0.8279 | 0.8871 |
> | VQ-Seg (Ours)| **0.6643** | **0.7852** | **0.8100** | **0.8507** | **0.9102** |
>
> VQ-Seg consistently outperforms all semi-supervised baselines across all settings, and even surpasses the fully supervised UNet using only 50% of the labeled data. These results demonstrate the strong scalability and efficiency of our method.
>
> > [W3] No external dataset is used to evaluate domain generalization.
>
> [A3] To evaluate external generalization, we conducted cross-domain experiments using the publicly available LUNA16 dataset [1], which also focuses on lung cancer segmentation. We compare our method and baselines trained on LC (5% labeled) and tested on LUNA16, along with a UNet trained directly on LUNA16 as an in-domain upper bound:
>
> | Method               | Training Data         | Testing Data | Dice↑  |
> |----------------------|------------------------|---------------|--------|
> | UNet (Full Sup.)     | LUNA16 (100% labeled)  | LUNA16        | 0.937  |
> | UNet (Full Sup.)     | LC (100% labeled)      | LUNA16        | 0.845  |
> | ABD (Semi-Sup.)      | LC (5% labeled)        | LUNA16        | 0.906  |
> | Unimatch (Semi-Sup.) | LC (5% labeled)        | LUNA16        | 0.913  |
> | VQ-Seg (Ours)    | LC (5% labeled)        | LUNA16        | **0.928** |
>
> Our method achieves the best generalization performance among all semi-supervised models and closely approaches the in-domain upper bound, demonstrating strong robustness to domain shift.
>
> > [W4] Concern over using DINOv2.
>
> [A4] We appreciate the reviewer’s insightful concern regarding the potential domain mismatch between DINOv2 and medical imaging tasks. While it is true that DINOv2 is pretrained on natural images, we selected it as our semantic teacher for several reasons:
>
> (1) Strong Generalization and Empirical Success in Medical Imaging:
> DINOv2 has demonstrated remarkable generalization capabilities across diverse downstream tasks. Despite being trained on natural images, it has shown strong transfer performance in medical domains. For example, [2] reports that DINOv2 achieves the best average segmentation accuracy among a range of pretraining methods. Additionally, [3] shows that directly using frozen DINOv2 features yields state-of-the-art results in medical image registration. These findings suggest that DINOv2 encodes rich and transferable semantic representations, even under domain shift.
>
> (2) Separation of Roles: DINOv2 for Quantization Alignment, DB for Domain Knowledge:
> DINOv2 is specifically used to align semantic representations before and after quantization, addressing the semantic degradation and information loss introduced by vector quantization. It serves as a stable, high-level anchor that preserves representation consistency across the quantization boundary. Crucially, medical domain knowledge is not derived from DINOv2; instead, it is injected via our Dual-Branch (DB) module, which incorporates task-specific features and medical semantics. This separation ensures that our model benefits from both robust general-purpose semantics (via DINOv2) and precise domain-specific guidance (via DB).
>
> (3) Comparison with Medical Foundation Models:
> To further validate our design choice, we conducted an ablation study replacing DINOv2 with several alternative foundation models, including ones pretrained on medical data. The Dice scores under 5% and 10% labeled data are shown below:
>
> | Foundation Model  | Dice (5% Labeled) | Dice (10% Labeled) |
> | ----------------- | ----------------- | ------------------ |
> | CLIP              | 0.6421            | 0.7483             |
> | BiomedCLIP        | 0.6507            | 0.7629             |
> | MAE               | 0.6386            | 0.7541             |
> | rad-DINO          | 0.6535            | 0.7793             |
> | DINOv2 (Ours) | **0.6643**        | **0.7852**         |
>
> Here, CLIP \[4] and BiomedCLIP \[5] are vision-language models trained on large-scale image–text datasets (natural and medical, respectively). MAE \[6] is trained via masked autoencoding, and rad-DINO \[7] follows a DINO-style architecture trained on radiology-specific images. As shown, DINOv2 consistently outperforms all alternatives across both labeling regimes, including models specialized for medical domains. These results empirically validate DINOv2 as a robust and effective semantic prior for semi-supervised medical image segmentation.
>
> > References:
>
> [1] Setio A A A, Traverso A, De Bel T, et al. Validation, comparison, and combination of algorithms for automatic detection of pulmonary nodules in computed tomography images: the LUNA16 challenge[J]. Medical image analysis, 2017, 42: 1-13.
>
> [2] Baharoon M, Qureshi W, Ouyang J, et al. Evaluating general purpose vision foundation models for medical image analysis: An experimental study of dinov2 on radiology benchmarks[J].
> arXiv preprint arXiv:2312.02366, 2023.
>
> [3] Song X, Xu X, Yan P. Dino-reg: General purpose image encoder for training-free multi-modal deformable medical image registration[C]//International Conference on Medical Image Computing and Computer-Assisted Intervention. Cham: Springer Nature Switzerland, 2024: 608-617.
>
> [4] Radford A, Kim J W, Hallacy C, et al. Learning transferable visual models from natural language supervision[C]//International conference on machine learning. PmLR, 2021: 8748-8763.
>
> [5] Zhang S, Xu Y, Usuyama N, et al. Biomedclip: a multimodal biomedical foundation model pretrained from fifteen million scientific image-text pairs[J]. arXiv preprint arXiv:2303.00915, 2023.
>
> [6] He K, Chen X, Xie S, et al. Masked autoencoders are scalable vision learners[C]//Proceedings of the IEEE/CVF conference on computer vision and pattern recognition. 2022: 16000-16009.
>
> [7] Perez-Garcia F, Sharma H, Bond-Taylor S, et al. Exploring scalable medical image encoders beyond text supervision[J]. Nature Machine Intelligence, 2025, 7(1): 119-130.
>
> ***
> Additional results/analyses on your concerns are in the revised manuscript. We hope these address your questions. Let us know if you need further discussion, experiments, or clarifications—happy to provide more details.

---

> > ### Author Response · Authors · 2025-08-03
> > **Looking forward to feedback**
> >
> > Dear Reviewer jSKd,
> >
> > Thank you very much for your valuable feedback and insights. We greatly appreciate the time and effort you have devoted to reviewing our work. We have carefully addressed the points you raised and would be grateful if you could review our responses to ensure that your questions and concerns have been fully addressed. We are more than willing to engage in further discussions.
> >
> > Warm regards,
> >
> > Authors of VQ-Seg

---

> > > ### Author Response · Authors · 2025-08-04
> > > **Looking forward to feedback**
> > >
> > > Dear Reviewer jSKd,
> > >
> > > Thank you very much for your valuable feedback and insights. We greatly appreciate the time and effort you have devoted to reviewing our work. We have carefully addressed the points you raised and would be grateful if you could review our responses to ensure that your questions and concerns have been fully addressed. We are more than willing to engage in further discussions.
> > >
> > > Warm regards,
> > >
> > > Authors of VQ-Seg

---

> > > > ### Author Response · Authors · 2025-08-05
> > > > **Looking forward to feedback**
> > > >
> > > > Dear Reviewer jSKd,
> > > >
> > > > Thank you very much for your valuable feedback and insights. We greatly appreciate the time and effort you have devoted to reviewing our work. We have carefully addressed the points you raised and would be grateful if you could review our responses to ensure that your questions and concerns have been fully addressed. We are more than willing to engage in further discussions.
> > > >
> > > > Warm regards,
> > > >
> > > > Authors of VQ-Seg

---

> > > > > ### Comment · Reviewer_jSKd · 2025-08-05
> > > > >
> > > > > I thank the authors for detailed rebuttal. My concerns are addressed and I am willing to increase my score.

---

> > > > > > ### Author Response · Authors · 2025-08-06
> > > > > >
> > > > > > Dear Reviewer jSKd,
> > > > > >
> > > > > > Thank you very much for your positive feedback. We are grateful that our response satisfactorily addressed your concerns.
> > > > > >
> > > > > > Please feel free to reach out if you have any further comments or questions.
> > > > > >
> > > > > > Warm regards,
> > > > > >
> > > > > > Authors of VQ-Seg

---

### Official Review · Reviewer_snt1 · 2025-07-01

**Clarity:** 3
**Significance:** 3
**Originality:** 3
**Rating:** 4
**Confidence:** 4

**Summary:**

This paper introduces VQ-Seg, a novel semi-supervised framework for medical image segmentation that leverages vector quantization (VQ) to address the limitations of traditional dropout-based consistency learning approaches. The authors propose three key innovations: (1) a Quantized Perturbation Module (QPM) that replaces dropout by shuffling codebook indices in the discrete VQ space, providing more stable and controllable perturbations; (2) a dual-branch architecture that jointly optimizes image reconstruction and segmentation tasks using shared post-VQ features to mitigate information loss from quantization; and (3) a Post-VQ Feature Adapter (PFA) that aligns quantized features with semantic embeddings from a foundation model to preserve high-level semantic information. The method is evaluated on a newly collected Lung Cancer (LC) dataset comprising 828 CT scans and the public ACDC dataset, demonstrating state-of-the-art performance across multiple metrics.

**Questions:**

1. How does the method prevent codebook collapse where only a subset of codewords are utilized?
2. What is the actual utilization rate of the 16384 codewords, and how does underutilization affect the perturbation space?
3. How sensitive is performance to the choice of foundation model？
4. How can you verify that shuffling spatial locations of codebook indices produces semantically valid perturbations for medical images, where spatial context is crucial for anatomical structures?

**Ethical Concerns:**

["NO or VERY MINOR ethics concerns only"]

**Final Justification:**

Considering that most of my concerns have been addressed, I will raise my score accordingly.

**Limitations:**

yes

**Paper Formatting Concerns:**

The paper formatting is suitable.

**Quality:**

3

**Strengths And Weaknesses:**

Strengths:
1. The replacement for dropout-based perturbation through vector quantization is well motivated, and supported by theoretical analysis showing that QPM offers bounded and numerically stable perturbations compared to the potentially unbounded KL divergence of high dropout rates.
2. The collection and annotation of 828 CT scans represents a significant contribution to the medical imaging community.

Weaknesses:
1. The relience on pre-trained foundation models for semantic guidance lacks novelty and the authors do not analyze the impact of different foundation models.
2. The experimental results lack error bars or statistical significance tests.
3. The paper sets codebook size following prior work, without exploring how this parameter affects the performance.
4. The choice of spatial shuffling over other perturbation strategies in the discrete space, e.g., nearest neighbor substitution, probabilistic replacement, lacks empirical justification.
5. The quality of the learned codebook is not analyzed. And how the learned codebook structure influences perturbation effectiveness is not explored.
6. The ablation study on the individual components should be provided beyond the incremental contributions.

---

> ### Author Rebuttal · Authors · 2025-07-30
>
> We sincerely appreciate your constructive feedback. Below, we provide point-by-point responses, with all revisions incorporated accordingly.
> ***
> > [W1] Limited novelty in using foundation models; lack of comparison.
>
> [A1] Rather than simply using a foundation model, we introduce a novel strategy that leverages it as a semantic anchor to calibrate quantized representations for semi-supervised medical image segmentation. The Post-VQ Feature Adapter (PFA) aligns discrete VQ features with high-level semantics, effectively mitigating quantization-induced degradation. As shown in the ablation study (Table 2), removing the PFA results in a performance drop: Dice decreases from 0.7852 to 0.7784, and Jaccard from 0.6731 to 0.6620.
>
> To evaluate the impact of different foundation models, we replace DINOv2 in the PFA module with several representative alternatives and report Dice scores under 5% and 10% labeled settings:
>
> | Foundation Model  | Dice (5% Labeled) | Dice (10% Labeled) |
> | ----------------- | ----------------- | ------------------ |
> | CLIP              | 0.6421            | 0.7483             |
> | BiomedCLIP        | 0.6507            | 0.7629             |
> | MAE               | 0.6386            | 0.7541             |
> | rad-DINO          | 0.6535            | 0.7793             |
> | DINOv2 (Ours) | **0.6643**        | **0.7852**         |
>
> CLIP \[1] and BiomedCLIP \[2] are vision-language models pre-trained on large-scale natural and medical image–text pairs, respectively. MAE \[3] is a vision model trained via masked image modeling. rad-DINO \[4] adopts a DINO-like architecture trained on radiology-specific datasets. As shown above, DINOv2 achieves the best performance across both settings, confirming its strong semantic capacity as the backbone for our PFA module.
>
> > [W2] No error bars or significance testing in experiments.
>
> [A2] Thank you for the constructive suggestion. To assess statistical robustness, we ran both Unimatch and VQ-SEG 10 times under the 5% labeled setting using different random seeds. The results are summarized below:
>
> | Method        | Dice (mean ± std) |
> | ------------- | ----------------- |
> | Unimatch      | 0.6491 ± 0.0124   |
> | VQ-SEG (Ours) | **0.6644 ± 0.0107**   |
>
> A paired t-test (p = 0.0013) and a Wilcoxon signed-rank test (p = 0.0021) confirm that the performance gain of VQ-SEG over Unimatch is statistically significant.
>
>
> > [W3] Codebook size not studied.
>
> [A3] Thank you for the helpful comment regarding codebook size.
> We conducted an ablation study to examine how the codebook size $K$ affects segmentation performance under 5% and 10% labeled settings:
>
> | Codebook Size | Dice (5%)  | Dice (10%) | Codebook Utilization (%) |
> | ------------- | ---------- | ---------- | --------------- |
> | 1024          | 0.6531     | 0.7608     | 100%            |
> | 2048          | 0.6582     | 0.7748     | 100%            |
> | 4096          | 0.6627     | 0.7775     | 99%             |
> | 16384         | **0.6643** | **0.7852** | 98%             |
> | 32768         | 0.6595     | 0.7764     | 95%             |
> | 65536         | 0.6415     | 0.7638     | 92%             |
>
> The results show that segmentation performance improves steadily as the codebook size increases, peaking at $K = 16384$, where Dice scores are highest and utilization remains high. Beyond this point, performance degrades and codebook utilization drops. This confirms that a codebook size of 16384 offers the best trade-off between representational capacity and generalization.
>
> > [W4] QPM lacks empirical justification vs. other strategies.
>
> [A4] Thank you for the insightful comment. Our Quantized Perturbation Module (QPM) adopts a probabilistic replacement strategy based on pairwise codeword distances (Eq. 4), rather than simple spatial shuffling.
>
> To justify this design, we compare QPM with two deterministic baselines under both 5% and 10% labeled settings:
>
> * Nearest Neighbor Replacement: each codeword is replaced with its closest neighbor.
> * Farthest Neighbor Replacement: each codeword is replaced with the most distant one.
>
> | Strategy          | Dice (5%)  | Dice (10%) |
> | ----------------- | ---------- | ---------- |
> | Nearest Neighbor  | 0.6591     | 0.7662     |
> | Farthest Neighbor | 0.6112     | 0.6931     |
> | QPM (proposed)    | **0.6643** | **0.7852** |
>
> QPM consistently achieves better performance than both baselines across different labeled settings.
>
> > [W5] No analysis of codebook quality or its impact on perturbation.
>
> [A5] Thank you for your comment. The quality of the learned codebook is analyzed in **[A3]**, where we examine how codebook size affects both segmentation performance and codeword utilization. We find that performance peaks when the codebook is sufficiently expressive and efficiently utilized (98% utilization at $K = 16384$). In contrast, excessively large codebooks suffer from underutilization and slightly reduced performance, while overly small codebooks, though fully utilized, lack diversity and representational capacity, leading to suboptimal results.
>
> This level of utilization is critical for QPM: a well-utilized codebook provides dense coverage of the latent space, ensuring that each replacement remains within a semantically meaningful neighborhood. Conversely, a sparse or underutilized codebook results in unstable and potentially harmful perturbations.
>
> > [W6] Missing ablation beyond incremental contributions.
>
> [A6] Thank you for your attention to the ablation analysis.  We would like to clarify that an ablation study on the individual components of our method is already included in the main paper (Table 2, Section 4.5). This table presents a step-by-step evaluation of the Quantized Perturbation Module (QPM), the dual-branch architecture (DB), and the Post-VQ Feature Adapter (PFA).
>
> The results show that each component contributes clear performance gains over the baseline, and their combination achieves the best overall performance. These findings confirm the effectiveness and complementarity of the proposed modules beyond mere incremental contributions. We hope this addresses the reviewer’s concern.
>
> > [Q1] Prevention of codebook collapse.
>
> [A7]  To clarify how our method prevents codebook collapse, we include an additional ablation study demonstrating that each component contributes to improved codebook utilization. The codebook size is set to 16,384.
>
> | Setting                | Dice    | Codebook Utilization (%) |
> |------------------------|---------|-----------------|
> | Base                   | 0.7443  | 11.6%           |
> | + QPM                  | 0.7701  | 87.2%           |
> | + QPM + DB             | 0.7784  | 90.1%           |
> | + QPM + DB + PFA       | **0.7852**  | **98.0%**       |
>
> Each component plays a distinct role in enhancing utilization. QPM introduces structured perturbations that activate a broader set of codewords. DB promotes representational diversity through joint reconstruction supervision. PFA encourages consistent and semantically meaningful codeword activation by aligning with high-level features. Together, these modules effectively mitigate codebook collapse.
>
> > [Q2] Utilization rate of 16384 codewords and its impact.
>
> [A8] Please refer to **[A3]** for detailed analysis. A codebook size of 16,384 provides the best trade-off between performance and utilization (98%).
>
> > [Q3] Sensitivity to foundation model choice.
>
> [A9] Please refer to **[A1]** for detailed analysis on the impact of different foundation models. DINOv2 consistently outperforms other alternatives, confirming its effectiveness.
>
> > [Q4] Validity of spatial shuffling in medical images.
>
> [A10] We appreciate the reviewer’s concern regarding the validity of spatial shuffling in medical images. Recent studies \[5] have shown that distances between codewords in the VQ codebook reflect semantic similarity. We observed a similar property in our experiments. Perturbations introduced by the QPM, which rely on controlled sampling of nearby codewords, help preserve anatomical semantics in the predicted segmentation. For instance, in lung cancer segmentation, QPM-perturbed predictions remain focused on cancer-relevant regions, maintaining their location and shape. In contrast, dropout often produces noisy outputs, including false positives in the background or mis-segmentation of surrounding structures such as the bronchi or chest wall. Although visualizations cannot be included at this stage due to NeurIPS policy, we will provide detailed qualitative and quantitative comparisons in the final version to substantiate this claim.
>
> ***
>
> > References:
>
> [1] Radford A, Kim J W, Hallacy C, et al. Learning transferable visual models from natural language supervision[C]//International conference on machine learning. PmLR, 2021: 8748-8763.
>
> [2] Zhang S, Xu Y, Usuyama N, et al. Biomedclip: a multimodal biomedical foundation model pretrained from fifteen million scientific image-text pairs[J]. arXiv preprint arXiv:2303.00915, 2023.
>
> [3] He K, Chen X, Xie S, et al. Masked autoencoders are scalable vision learners[C]//Proceedings of the IEEE/CVF conference on computer vision and pattern recognition. 2022: 16000-16009.
>
> [4] Perez-Garcia F, Sharma H, Bond-Taylor S, et al. Exploring scalable medical image encoders beyond text supervision[J]. Nature Machine Intelligence, 2025, 7(1): 119-130.
>
> [5] Hu T, Zhang J, Yi R, et al. Improving autoregressive visual generation with cluster-oriented token prediction[C]//Proceedings of the Computer Vision and Pattern Recognition Conference. 2025: 9351-9360.
>
> ***
>
> Additional results/analyses on your concerns are in the revised manuscript. We hope these address your questions. Let us know if you need further discussion, experiments, or clarifications—happy to provide more details.

---

> > ### Comment · Reviewer_snt1 · 2025-08-05
> >
> > Thank the reviewers for their effort and time. Most of my concerns have been addressed, and I greatly appreciate it. I strongly recommend including the complete results, including the error bars, in the final version. I will raise my score accordingly.

---

> ### Author Response · Authors · 2025-08-03
> **Looking forward to feedback**
>
> Dear Reviewer snt1,
>
> Thank you very much for your valuable feedback and insights. We greatly appreciate the time and effort you have devoted to reviewing our work. We have carefully addressed the points you raised and would be grateful if you could review our responses to ensure that your questions and concerns have been fully addressed. We are more than willing to engage in further discussions.
>
> Warm regards,
>
> Authors of VQ-Seg

---

> > ### Author Response · Authors · 2025-08-04
> > **Looking forward to feedback**
> >
> > Dear Reviewer snt1,
> >
> > Thank you very much for your valuable feedback and insights. We greatly appreciate the time and effort you have devoted to reviewing our work. We have carefully addressed the points you raised and would be grateful if you could review our responses to ensure that your questions and concerns have been fully addressed. We are more than willing to engage in further discussions.
> >
> > Warm regards,
> >
> > Authors of VQ-Seg

---

### Official Review · Reviewer_4i9r · 2025-07-02

**Clarity:** 3
**Significance:** 2
**Originality:** 3
**Rating:** 4
**Confidence:** 3

**Summary:**

This paper presents VQ-seg, a novel semi-supervised medical image segmentation framework. It uses Quantized Perturbation Module (QPM) for robust representations in VQ feature space and a dual-branch architecture with Post-VQ Feature Adapter (PFA) for high-level semantic integration. Experiments on LC and ACDC datasets show the performance compared to previous work.

**Questions:**

Please see weaknesses.

**Ethical Concerns:**

["NO or VERY MINOR ethics concerns only"]

**Final Justification:**

I appreciate the efforts by the authors for rebuttal. The rebuttal partially addresses my concerns, but there is still room for this paper to improve. After reading other reviews and authors' response, I choose to keep my original rating.

**Limitations:**

Yes

**Quality:**

3

**Strengths And Weaknesses:**

Strengths:

1.	The proposed dataset large-scale Lung Cancer (LC) dataset is valuable.

2.	The paper provides theoretical analysis which is good.

3.	The experimental results show the advantages of the proposed method compared to previous work.

Weaknesses:

1.	Table 1 only shows the results on the LC dataset with two labeled ratio settings 5% and 10%. More labeled ratio settings are suggested such that it can show the trend of the performance with increasing ratio of labeled data.

2.	In Sec. 5, it mentions that “However, the effectiveness of VQ-seg depends on the quality and relevance of the foundation model, which may introduce bias if not carefully selected and validated.” Can the authors provide more details about this limitation? What is the impact of different choices of foundation models? Please provide more discussion about the limitations.

3.	It claims that “QPM enables a more structured and controllable mechanism for representation perturbation by shuffling the spatial locations of codebook indices, offering enhanced interpretability and stability compared to traditional dropout.” However, there is no evidence, such as the visualization of the perturbation generated by QPM versus traditional dropout. Please add visualization results.

---

> ### Author Rebuttal · Authors · 2025-07-30
>
> We sincerely appreciate your constructive feedback. Below, we provide point-by-point responses, with all revisions incorporated accordingly.
> ***
> > [W1] Limited labeled ratio settings.
>
> [A1] We agree that incorporating additional labeled ratio settings can better illustrate the performance trend under varying levels of supervision. In response, we have extended our experiments on the LC dataset to include additional labeled ratios of 20%, 50%, and 100%. The updated results, evaluated using the Dice coefficient, are summarized below:
>
> | Method         | 5%     | 10%    | 20%    | 50%    | 100%   |
> |----------------|--------|--------|--------|--------|--------|
> | UNet-S    | 0.4343 | 0.6490 | 0.7205 | 0.7880 | 0.8345 |
> | MCNet          | 0.6378 | 0.7555 | 0.7812 | 0.8203 | 0.8751 |
> | ABD            | 0.6414 | 0.7468 | 0.7780 | 0.8235 | 0.8824 |
> | Unimatch       | 0.6493 | 0.7511 | 0.7855 | 0.8279 | 0.8871 |
> | VQ-Seg (Ours) | **0.6643** | **0.7852** | **0.8100** | **0.8507** | **0.9102** |
>
> As shown above, VQ-Seg consistently outperforms all baselines across different supervision levels. Notably, with only 50% labeled data, our method already surpasses the fully supervised UNet baseline (0.8345). The consistent improvements as more annotations are available further highlight the scalability and effectiveness of our approach.
>
>
> > [W2] Lack of discussion on the impact of different foundation models.
>
> [A2] Thank you for the helpful comment. We provide further clarification below.
>
> We assess the impact of different foundation models by replacing DINOv2 in VQ-Seg and reporting Dice scores under the 5% and 10% labeled settings:
>
> | Foundation Model  | Dice (5% Labeled) | Dice (10% Labeled) |
> | ----------------- | ----------------- | ------------------ |
> | CLIP              | 0.6421            | 0.7483             |
> | BiomedCLIP        | 0.6507            | 0.7629             |
> | MAE               | 0.6386            | 0.7541             |
> | rad-DINO          | 0.6535            | 0.7793             |
> | DINOv2 (Ours) | **0.6643**        | **0.7852**         |
>
> CLIP [1] and BiomedCLIP [2] are vision-language models pre-trained on large-scale natural and medical image–text pairs, respectively. MAE [3] is a vision model pre-trained via masked image modeling, while rad-DINO [4] employs a DINO-style architecture trained on radiology-specific datasets.
>
> As shown above, DINOv2 consistently achieves the best performance across both supervision levels. It surpasses CLIP and MAE by more than 2% in absolute Dice score and outperforms BiomedCLIP and rad-DINO by a non-trivial margin.
>
> These results reveal that foundation models pretrained with self-supervised, vision-only objectives (e.g., DINOv2, rad-DINO) produce more structure-aware representations and outperform vision-language models (e.g., CLIP, BiomedCLIP) on segmentation tasks. Among all candidates, DINOv2, which is used in our method, achieves the highest overall performance.
>
> > [W3] Lack of evidence supporting the benefits of QPM over dropout.
>
> [A3] Thank you for the insightful comment. Due to NeurIPS rebuttal policy, we are unable to modify the main paper or include additional figures at this stage. However, we would like to clarify that the perturbations introduced by QPM are anatomically meaningful. For example, in lung cancer segmentation, QPM-perturbed predictions remain focused on cancer-related regions. In contrast, dropout often introduces noisy outputs, including false positives in the background or mis-segmentation of adjacent structures such as bronchi or the chest wall. This behavior reflects the controlled nature of QPM, which samples from semantically similar codewords and thereby preserves anatomical semantics in the predictions. Although visualizations cannot be included at this stage, we will provide detailed qualitative and quantitative comparisons in the final version to support this claim.
>
> ***
> > References:
>
> [1] Radford A, Kim J W, Hallacy C, et al. Learning transferable visual models from natural language supervision[C]//International conference on machine learning. PmLR, 2021: 8748-8763.
>
> [2] Zhang S, Xu Y, Usuyama N, et al. Biomedclip: a multimodal biomedical foundation model pretrained from fifteen million scientific image-text pairs[J]. arXiv preprint arXiv:2303.00915, 2023.
>
> [3] He K, Chen X, Xie S, et al. Masked autoencoders are scalable vision learners[C]//Proceedings of the IEEE/CVF conference on computer vision and pattern recognition. 2022: 16000-16009.
>
> [4] Perez-Garcia F, Sharma H, Bond-Taylor S, et al. Exploring scalable medical image encoders beyond text supervision[J]. Nature Machine Intelligence, 2025, 7(1): 119-130.
>
> ***
> We have incorporated additional results and detailed analyses addressing your concerns into the revised version of the manuscript. We hope these supplements adequately address your questions and improve the clarity and robustness of our work. Please do not hesitate to let us know if you require further discussion, additional experiments, or any clarifications—we are happy to provide more details to address your concerns comprehensively.

---

### Decision · Program_Chairs · 2025-09-17

**Decision:**

Accept (poster)

**Comment:**

This paper introduces VQ-Seg, a semi-supervised segmentation framework that replaces dropout-based consistency learning with a vector-quantization-based Quantized Perturbation Module. The framework further includes a dual-branch design and Post-VQ Feature Adapter (PFA) guided by a foundation model. The paper also contributes a new annotated LC dataset.

Reviewers initially expressed concerns about dataset scale, limited labeled ratios, foundation model dependence, codebook justification, and lack of error bars. However, the authors provided extensive new experiments in their rebuttal addressing many concerns with quantitative evidence. Three of the four reviewers raised their scores to weak accept and all reviewers now give positive ratings indicating unanimous support for acceptance.

Reviews and AC encourage the authors to include their responses in the camera-ready version, including but not limited to:

1. Include all additional experimental results
2. Add error bars and statistical significance tests to main results
3. Add t-SNE visualizations of codebook evolution
4. Expand discussion on foundation model selection